# Exploring effector protein dynamics and natural fungicidal potential in rice blast pathogen *Magnaporthe oryzae*

Jannatul Ferdousy[1]☯, Tanjin Barketullah Robin[2]☯, Mst Sanjida Nasrin[3]☯*, Istiak Ahmed[4]☯, Tawsif Hossain[5]☯, Md Mehedi Hasan[6]☯, Mehrab Hassan Soaeb[7]☯, Md. Ahsanul Tamim[6]☯, Nusrat Jahan Yeasmin[8]☯, Ummay Habiba[6]☯, Nadim Ahmed[2]☯, Nurul Amin Rani[2]☯, Md Shishir Bhuyian[2]☯, Suvarna N. Vakare[9], Abu Tayab Moin[10]☯*, Rajesh B. Patil[9]*, Mohammad Shahadat Hossain[11]*

1 Plant Breeding Division, Bangladesh Rice Research Institute, Gazipur, Bangladesh, 2 Faculty of Biotechnology and Genetic Engineering, Sylhet Agricultural University, Sylhet, Bangladesh, 3 Department of Bioinformatics, School of Biosciences, University of Skövde, Skövde, Sweden, 4 Department of Pharmacy, Faculty of Science, East West University, Dhaka, Bangladesh, 5 Department of Microbiology, Primeasia University, Dhaka, Bangladesh, 6 Department of Biotechnology, Faculty of Agriculture, Bangladesh Agricultural University, Mymensingh, Bangladesh, 7 Deparment of Agriculture, Faculty of Agriculture, Hajee Mohammad Danesh Science and Technology University, Dinajpur, Bangladesh, 8 Department of Soil Science, Faculty of Agriculture, Bangladesh Agricultural University, Mymensingh, Bangladesh, 9 Department of Pharmaceutical Chemistry, Sinhgad Technical Education Society's, Sinhgad College of Pharmacy, Pune, Maharashtra, India, 10 Department of Genetic Engineering and Biotechnology, Faculty of Biological Sciences, University of Chittagong, Chattogram, Bangladesh, 11 Department of Computer Science and Engineering, University of Chittagong, Chattogram, Bangladesh

☯ These authors contributed equally to this work.
* rajesh.patil@sinhgad.edu (RBP); hossain_ms@cu.ac.bd (MSH); tayabmoin786@gmail.com (ATM); sanjida5260@gmail.com (MSN)

**Data Availability Statement:** All relevant data are within the paper and its Supporting Information files.

## Abstract

Rice blast, caused by *Magnaporthe oryzae*, is one of the most destructive fungal diseases in rice, resulting in major economic losses worldwide. Genetic and genomic studies have identified key genes and proteins, such as AvrPik variants and MAX proteins, that are crucial for the pathogen's virulence. These effector proteins interact with specific alleles of the Pik gene family on rice chromosome 11, modulating the host's immune response. In this study, we investigated 35 plant-derived metabolites known for their antifungal properties as potential fungicides against *M. oryzae*. Using molecular docking, we identified Hecogenin and Cucurbitacin E as strong binders to MAX40 and APIKL2A proteins, which are essential for the fungus's immune evasion and pathogenicity. Molecular dynamics simulations further confirmed that these compounds form stable, strong interactions with the target proteins, validating their potential as therapeutic agents. Additionally, the compounds were evaluated based on Lipinski's rule of five and toxicity predictions, indicating their suitability for agricultural use. These results suggest that Hecogenin and Cucurbitacin E could serve as promising lead candidates in the development of novel fungicides for rice blast, offering new strategies for crop protection and sustainable agricultural practices.

**Funding:** The author(s) received no specific funding for this work.

**Competing interests:** The authors have declared that no competing interests exist.

## 1. Introduction

Rice blast disease, caused by the fungus *Magnaporthe oryzae*, is a major threat to rice production globally, especially in tropical and subtropical regions where the warm and humid climate promotes fungal growth [1, 2]. This disease results in substantial economic losses, impacting crop yield, farmer income, and increasing production costs due to the reliance on fungicides and other control measures [3, 4] Moreover, rice blast disease diminishes rice's market value, placing additional financial pressure on farmers [5]. Given the critical role of rice in the agriculture and economies of Asia, managing rice blast disease is essential, especially in key rice-producing nations like China and India, which together account for over half of global rice production [6, 7]. In Bangladesh, rice remains a central food source and an economic mainstay, contributing 16% of the national GDP and 70% of the agricultural GDP [8].

The high genetic variability of *M. oryzae* makes managing rice blast disease particularly challenging [9]. The fungus can rapidly evolve new strains that overcome resistance genes in rice, causing frequent resistance breakdowns [10]. Environmental dependency further complicates disease management, as outbreaks are more severe in warm, humid conditions, complicating preventive strategies [11]. Although chemical fungicides are commonly used to control the disease, they pose environmental and health risks, prompting the exploration of safer, sustainable alternatives like plant-derived compounds with natural antifungal properties [12].

*M. oryzae* secretes various effector proteins, including AvrPik variants and MAX proteins, during infection to evade and suppress the rice plant's immune defenses [13]. These effectors act as molecular disruptors within the rice cells, altering signaling pathways essential for immune response and cell wall integrity [14]. Many effector proteins share structural motifs that enable their manipulation of host cell functions, though some exhibit unique structural differences, providing alternative therapeutic targets. Targeting structurally diverse effectors could reduce the pathogen's ability to develop resistance, as it would require multiple simultaneous mutations. Identifying structurally distinct effector proteins and pairing them with various plant-derived compounds could help mitigate resistance development and sustain disease management efficacy.

This study investigates the potential of plant-derived secondary metabolites to target a range of *M. oryzae* effector proteins, using a combination of molecular docking and molecular dynamics simulations to evaluate binding affinities and interaction profiles. Among the compounds studied, Hecogenin and Cucurbitacin E emerged as particularly promising due to their distinctive biological properties and antifungal activity. Hecogenin, a steroidal saponin primarily derived from *Agave* species like *Agave sisalana* and *Agave aurea*, exhibits a range of pharmacological activities, including anticancer, antifungal, and hypotensive effects. It is a selective inhibitor of human UDP-glucuronosyltransferases (UGTs), enzymes involved in the detoxification of drugs and toxins. Hecogenin's inhibitory action on UGTs underscores its potential to enhance drug metabolism and overcome drug resistance, while its antifungal properties align well with its role as a fungicidal agent in this study against *M. oryzae* [15]. The compound's multifaceted biological effects and ability to influence key enzymatic pathways make it a valuable candidate in both drug design and agricultural research.

Cucurbitacin E, a tetracyclic triterpenoid predominantly found in cucurbitaceous plants such as pumpkins, gourds, and cucumbers, is well-known for its wide array of biological activities, including antitumor, hepatoprotective, anti-inflammatory, and antioxidant effects. It has shown efficacy against cancer cells and protective effects on liver function, establishing its potential for therapeutic applications in oncology and liver disease. Cucurbitacin E also exhibits anti-inflammatory properties by suppressing pathways such as NF-κB, a central regulator of immune responses and inflammation [16, 17]. In the context of this study, Cucurbitacin E's

ability to target key effector proteins involved in *M. oryzae* infection further supports its potential as a novel antifungal agent.

Our results indicate that Hecogenin and Cucurbitacin E demonstrate strong binding affinities across different effector proteins, with variable interaction profiles influenced by the structural characteristics of each target protein. This finding supports their potential as lead compounds in developing dual-targeted fungicidal formulations that may reduce resistance development by *M. oryzae*. Additionally, our study assessed the fungicide-like properties of these metabolites according to Lipinski's rule of five and evaluated their toxicity, offering a comprehensive overview of their viability as antifungal agents. These results underscore the potential of combining targeted effector inhibition with natural antifungal compounds, opening new avenues for sustainable rice blast disease management.

However, the molecular docking approach has inherent limitations. It relies on static structural models and scoring algorithms that may not accurately reflect the dynamic and complex nature of protein-ligand interactions in physiological conditions. Furthermore, the lack of experimental confirmation restricts the ability to evaluate the true binding affinity, bioavailability, and effectiveness of these compounds in real biological systems. While our findings provide a promising foundation, further *in vitro* and *in vivo* studies are essential to validate the efficacy and safety of these compounds for practical applications

## 2. Methods

The stepwise methods used in the study are illustrated in Fig 1.

### 2.1. Preparation of the target proteins

Effector proteins of *M. oryzae*, including AvrPik variants and MAX proteins, were selected based on an extensive literature review. The structures of these proteins were obtained from The RCSB Protein Data Bank [18, 19]. Subsequently, the target protein structures were refined using the BIOVIA Discovery Studio Visualizer program to remove undesired ligands, metals, and ions [20]. For protein visualization, UCSF Chimera Software was used.

### 2.2. Ligand preparation

Thirty secondary metabolites with established antifungal activity from various plant sources were selected through a comprehensive literature review. These substances and inhibitors are potential therapeutic candidates against fungi. Evaluation of their binding affinities utilized reference ligands like strobilurin. The structures of these metabolites, available in SDF (3D) format, were retrieved from the PubChem database [21] and converted to PDB format using Open Babel v2.3, a versatile chemical data tool supporting over 110 different formats [22], to prepare them for further analysis.

### 2.3. Molecular docking and binding interaction analysis

A flexible docking methodology was employed to investigate the binding affinity between metabolites and proteins [23]. The molecular docking of the selected ligands with the target proteins was conducted using AutoDock Vina from the Python Prescription 0.8 (PyRx) package [24]. Prior to docking, all compounds and reference drugs underwent minimization and conversion to PDBQT format. The docking procedure itself was executed using Vina Wizard. For the analysis of binding sites and visualization of results, PyMOL v2.0 was utilized to identify polar and non-polar residues and examine the binding locations of specific metabolites [25, 26].

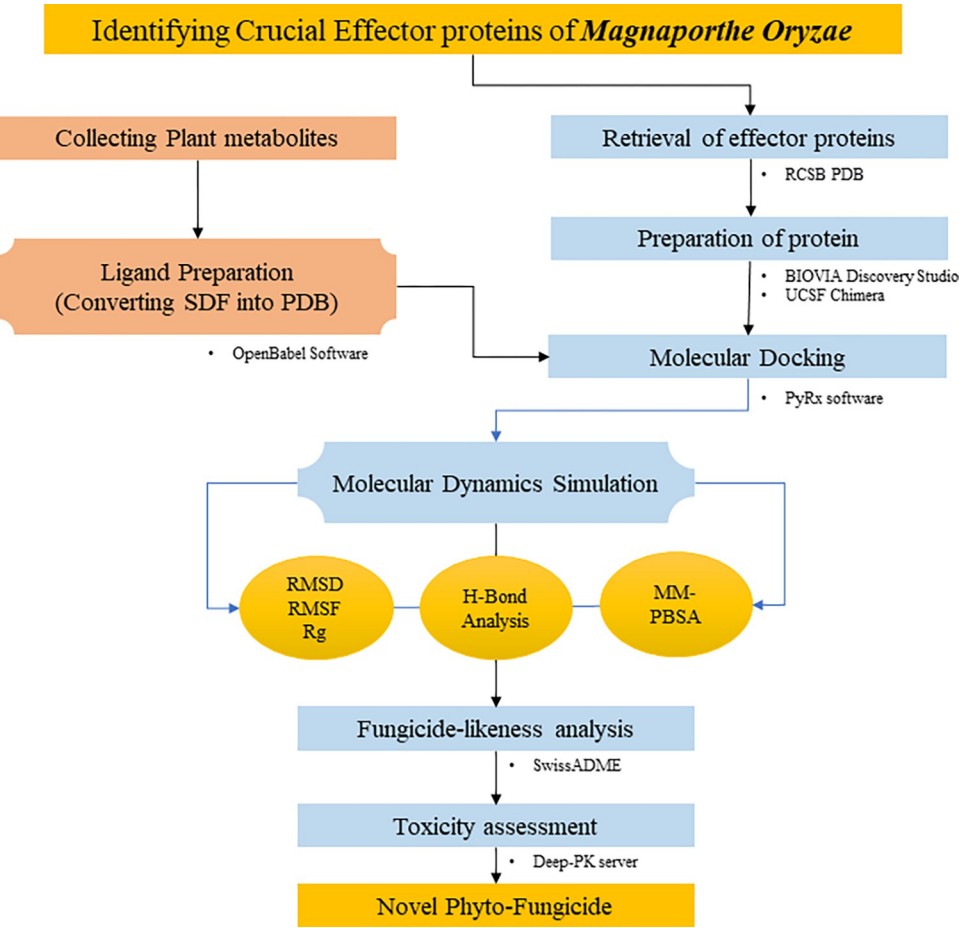

**Fig 1. Step-by-step flowchart detailing the methods used in this study.**

## 2.4. Molecular dynamics simulation analyses

Deeper insights into the binding affinities and possible intricate stabilizing properties of Hecogenine (HEC) and Strobulirin (STR) were obtained from the molecular dynamics (MD) simulations selecting the respective docked complexes of these ligands with rice effector proteins namely ApikL-2a, ApikL-2F, AVR-Pia, AVR-Pib, AVR-Pii, AVR-PikA, AVR-PikC, AVR-PikD, AVR-PikE, AVR-PikF, AVR-PizT, MAX47, MAX60, MAX67. The 100 ns MD simulations were conducted with the Gromacs-2020.4 [27, 28] program on the HPC cluster at Bioinformatics Resources and Applications Facility (BRAF), C-DAC, Pune. The topologies of proteins were built using the parameters from the CHARMM-36 force field [29, 30], while ligand topologies were obtained from the cGenFF server [30]. The respective complexes were held in a dodecahedron unit cell such that the edges of the system were 1 nm away from the edges of the box. Subsequently, the systems were solvated with water molecules employing the TIP3P water model [31] and neutralized by adding sodium and chloride counter-ions to achieve 0.15 mole concentration. Energy minimization was performed to relieve the steric strains with the steepest descent algorithm until the force constant reached the 100 kJ mol-1 nm-1 threshold. The systems were then equilibrated at constant temperature and pressure conditions, NVT and NPT conditions, at 300 K temperature and 1 atm pressure condition using a modified Berendsen thermostat [32] and barostat [33], respectively, for 1 ns each. The

100 ns MD simulations were performed on the equilibrated systems where the temperature conditions of 300 K were achieved with a modified Berendensen thermostat, and pressure conditions of 1 atm were achieved with the Parrinello-Rahman barostat [34]. The covalent bonds in the systems were restrained with the LINCS algorithm [35]. The long-range electrostatic energies were computed with the Particle Mesh Ewald (PME) method [36] at a cut-off of 1.2 nm. After removing the periodic boundary conditions, the trajectories of each complex were analyzed for the root mean square deviations (RMSD) in the backbone atoms of proteins. The RMSD in ligand atoms was analyzed by selecting protein backbone atoms for the most minor square fitting and ligand atoms for RMSD calculations, effectively showing the relative positions of ligands compared to initial position during the simulation.

Further, the root mean square fluctuation (RMSF) in the side chain atoms was analyzed. The radius of gyration (Rg) of the protein structure from its center of mass was analyzed to investigate the stability and compactness of the corresponding systems. During the simulation, the hydrogen bonds between the ligand and the protein residues were analyzed and visually inspected in the trajectories extracted at steps 0, 25, 50, 75, and 100 ns. Molecular Mechanics General Born surface area and surface area solvation (MM/GBSA) [37] calculations were performed on the trajectories isolated from the simulation period 75 to 100 ns at each 100 ps time step. The entropic energies were accounted for to obtain each complex's binding free energies (ΔGbinding kcal/mol). The protein-ligand structures were rendered in PyMOL [38] and graphs were obtained from XMGRACE [39].

## 2.5. Fungicide-like properties and toxicity analysis

The fungicide-likeness of natural compounds was assessed using Lipinski's rule of 5, a well-established criterion for drug-likeness, given the absence of similar criteria for fungicides [40]. The SwissADME server (http://www.swissadme.ch/) was utilized to evaluate the fungicidal qualities of the top metabolites [41, 42], providing predictions based on the compounds' characteristics related to Lipinski's Rule [42, 43]. Afterwards, the general toxicity of the identified compounds was forecasted using the pkCSM web-based server (https://biosig.lab.uq.edu.au/pkcsm/) [44].

# 3. Result

## 3.1. Preparation of the target proteins

Through an extensive literature review, fourteen crucial effector proteins necessary to inhibit M. oryzae, the causative agent of rice blast disease, were identified. These proteins, including APIKL2A, APIKL2F, AVRPIA, AVRPIB, AVRPII, AVRPIKA, AVRPIKC, AVRPIKD, AVRPIKE, AVRPIKF, AVRPIZT, MAX60, MAX47, and MAX67, were sourced from the Protein Data Bank with the following PDB IDs: 7NLJ, 6FUD, 7QPX, 7BNT, 6G11, 7B1I, 6Q76, 5Z1V, 7PP2, 2LW6, 7NMM, 72JY, 7ZK0, 7ZKD respectively. Afterwards, The obtained proteins were processed using BIOVIA Discovery Studio to remove undesired macromolecule ligands, water molecules, and heteroatoms from their structures. Subsequently, molecular graphics and analyses were conducted to explore their properties and potential for inhibiting M. oryzae. The proteins were visualized using PyMOL software, as depicted in Fig 2.

## 3.2. Ligand preparation

A thorough literature review was conducted to compile a list of 35 metabolites (S1 Table) derived from diverse plant species, recognized for their antifungal properties. These metabolites, including inhibitors and chemicals, hold promise as therapeutic agents for combating various fungal infections. 3D structures of these 30 metabolites were obtained from the

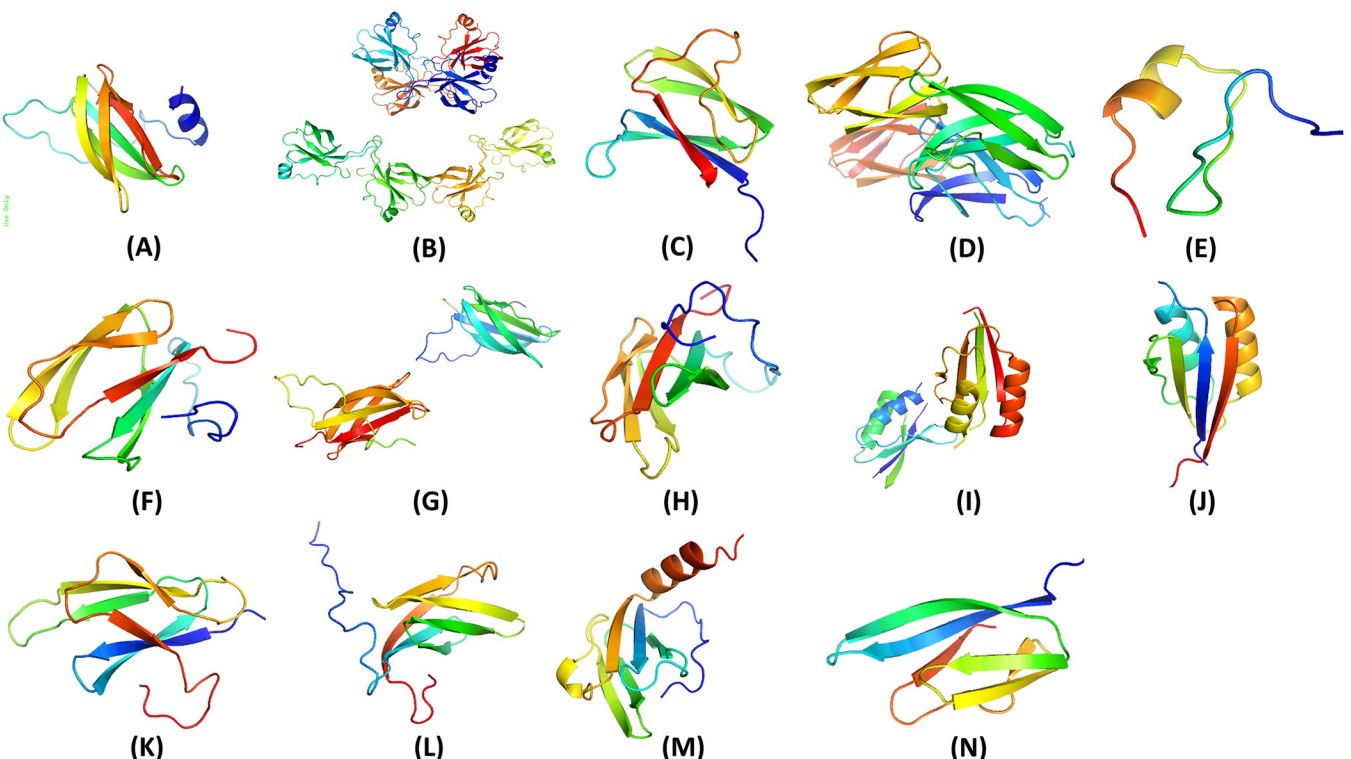

**Fig 2.** Tertiary structure of A) APIKL2A, B) APIKL2F, C) AVRPIA, D) AVRPIB, E) AVRPII, F) AVRPIKA, G) AVRPIKC, H) AVRPIKD, I) AVRPIKE, J) AVRPIKF, K) AVRPIZT, L) MAX60, M) MAX47, N) MAX67.

PubChem database in SDF format, and subsequently converted to PDB format using Open Babel v2.3 software. The same method was applied to convert the reference compound strobilurin into PDB format.

## 3.3. Molecular docking studies and binding site analysis

Molecular docking experiments were performed using 14 effector proteins as receptors and 35 plant metabolites (S2 Table) as ligands. Notably, Hecogenin and Cucurbitacin E exhibited the highest and most consistent binding affinities across all proteins. Hecogenin achieved the best docking energy of -8.8 kj/mol with protein APIKL2A, binding to residues LYS29, TYR25, ILE108, LYS109, and GLY92. Meanwhile, Cucurbitacin E demonstrated a maximum docking energy of -8.0 kj/mol with MAX40 protein, interacting with residues ARG-65, TRP-54, CYS-108, and GLY-106. Despite showing varied energy levels with some proteins, Cucurbitacin E consistently displayed superior binding energy compared to other metabolites. Several metabolites were excluded due to altered drug effects and toxicity. Table 1 illustrates that Hecogenin and Cucurbitacin E outperformed the reference metabolite Strobilurin. The binding sites, crucial for complex formation, are depicted in Figs 3–5 and S1, S2 Figs. Among Hecogenin and Cucurbitacin E, Hecogenin showed to have Pi-alkyl bond more with all the proteins, whereas Cucurbitacin E showed to have more strong hydrogen bond with most of the proteins.

## 3.4. Molecular dynamics simulation studies

**3.4.1. Root mean square deviation.** The results of RMSD in the backbone atoms of rice effector proteins showed that the proteins ApikL-2a, ApikL-2f, AVR-Pib, AVR-PikA,

**Table 1. Binding energy and binding site analysis of best complexes.**

| Ligand | Protein | Residues involved in interactions | Binding Energy |
|---|---|---|---|
| Hecogenin | APikl2a | LYS29, TYR25 ILE108, LYS109, GLY92 | -8.8 |
| | apikl2f | LEU95, ARG94, LEU106, TYR25 | -8.5 |
| | AVR_pikF | GLU67 | -6.5 |
| | AVR-pia | ASN72, LEU70, PHE57, PRO77 | -7.7 |
| | AVRPib | PHE44, TYR43, LYS42, ARG23, GLU22 | -8.1 |
| | AVR-pii | LYS50, TYR56, ASN70 | -6.1 |
| | AVR-pikA | LYS109, LYS108, LEU106, LEU35, ILE33 | -7.5 |
| | AVR-pikC | PRO111, LYS109, LYS108, LEU35 | -8.3 |
| | AVR-pikD | PHE44, HIS46, PRO52, HIS86, LYS75, TRP74 | -7.8 |
| | AVR-PikE | LEU209, LYS248, ALA244, SER212, THR213, LYS240 | -7.2 |
| | AVR-Piz-t | ARG32, TYR29, TRR34 | -6.7 |
| | MAX47 | TRP54, GLU60, ALA62, LYS57, ARG65 | -7.9 |
| | MAX60 | HIS25, MET22, VAL49, PRO20, HIS21 | -8.3 |
| | MAX67 | TYR49, LYS40 | -6.3 |
| Cucurbitacin E | APikl2a | ARG-94, GLY-92, GLY-107, ASN-23, TYR-25 | -7.4 |
| | apikl2f | ASN-23, SER-96, GLY-97, ILE-99, LEU-104, TYR-62 | -7.2 |
| | AVR_pikF | LYS-75 | -5.9 |
| | AVR-pia | GLN-75, PRO-77, GLU-58 | -6.7 |
| | AVRPib | TRP-26, THR-13, PRO-30, ARG-35 | -7.5 |
| | AVR-pii | ASN-70, GLY-55, LYS-68 | -5.4 |
| | AVR-pikA | PHE-44, THR-69, TRP-56 | -7.3 |
| | AVR-pikC | THR-69, PHE-44 | -7.2 |
| | AVR-pikD | ASP-77, THR-69 | -6.8 |
| | AVR-PikE | THR-213, SER-212, LEU-209, ALA-244 LYS-248 | -6.5 |
| | AVR-Piz-t | TYR-29, TRP-34, GLY-35, THR-36, ARG-46, LYS-51 | -6.7 |
| | MAX47 | ARG-65, TRP-54, CYS-108, GLY-106 | -8.0 |
| | MAX60 | ARG-37, ILE-35, TYR-94, VAL-48, LYS-24 | -7.9 |
| | MAX67 | TYR49, THR47, ASP46, ASN43, LYS40 | -5.8 |
| Strobilurin (Control) | Apikl2A | GLY97, ILE99, LEU95, GLU24, ILE105, LEU104, TYR62 | -7.2 |
| | Apikl2F | ARG94, LEU35, TYR25, LYS29, ILE26 | -6.5 |
| | Avrpia | LEU70, PHE57, GLU56, HIS55 | -6.2 |
| | AvrPib | PRO30, ARG35, VAL27, PHE37, TRP26, ARG40, ILE24, LYS19 | -7.1 |
| | Avrpii | LYS50, GLY57, GLY55, TYR56, CYS54, ALA63 | -5.4 |
| | AvrpikA | TRP74, LYS75, TYR71, PRO52, THR69, ILE49 | -6.5 |
| | AvrpikC | PRO50, TYR71, TRP74, MET76 | -6.6 |
| | AvrpikD | TRP74, TYR71, THR69, ILE49, HIS46 | -6.8 |
| | AvrpikE | LYS248, VAL249, LYS205, ALA208, THR213, LEU209, LEU241 | -5.7 |
| | Avrpikf | VAL56, ALA52, ALA53, LEU68, VAL50, VAL71 | -5.4 |
| | Avrpizt | LYS51, GLU50, TRP34, TYR29, ILE37 | -6.5 |
| | Max47 | GLN61, ALA62, TRP63, ARG65, SER55, TRP54 | -6.9 |
| | Max60 | TYR94, ILE35, VAL48, ILE47, GLY23, MET22 | -6.9 |
| | Max67 | ALA42, SER41, LEU48, LYS40, TYR49, VAL38, TYR29 | -5.4 |

AVR-PikC, and MAX67 stabilized reasonably, and the RMSD converged stably throughout the simulation with RMSD below 0.2 nm (Fig 6). However, ApikL-2f in complex with STR showed significant deviations after around 80 ns, reaching RMSD of a maximum of 0.3 nm. In

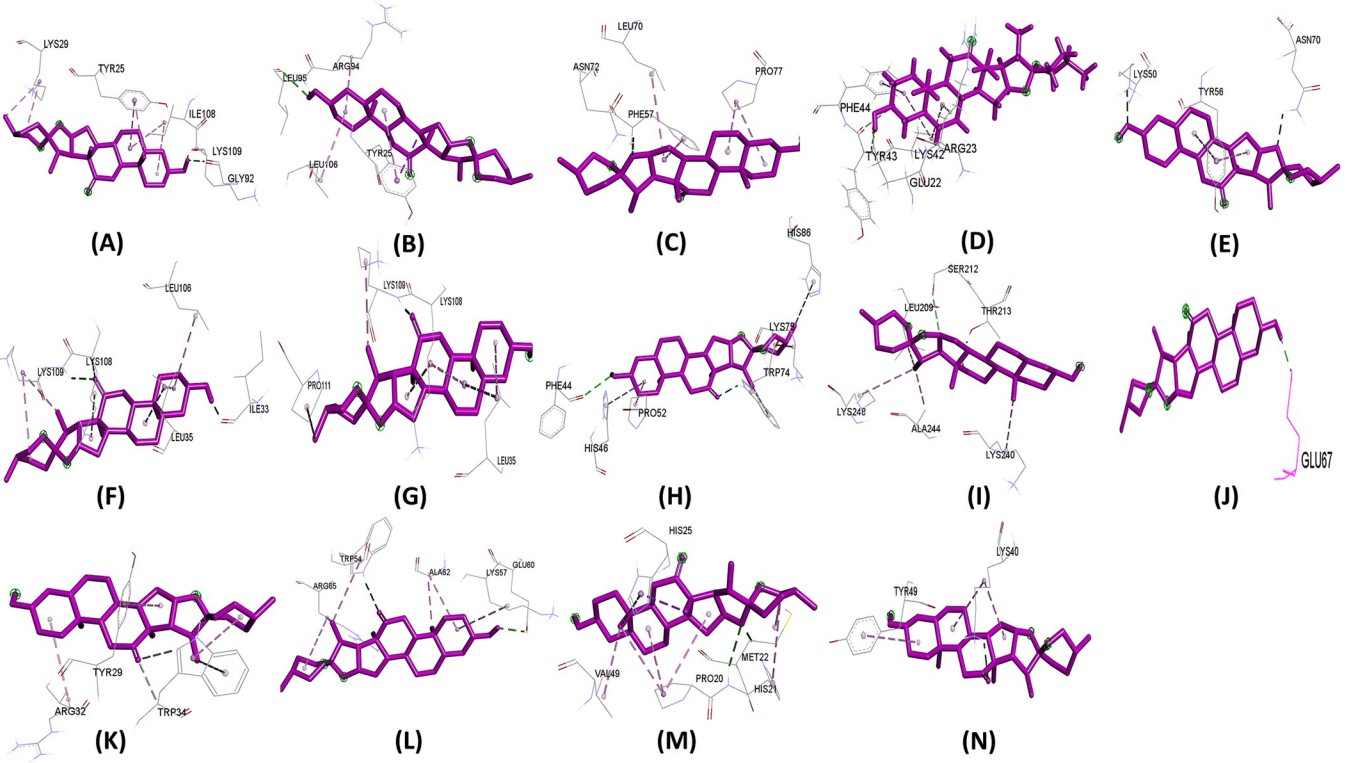

**Fig 3.** Binding site of ligand Hecogenin with protein A) APIKL2A, B) APIKL2F, C) AVRPIA, D) AVRPIB, E) AVRPII, F) AVRPIKA, G) AVRPIKC, H) AVRPIKD, I) AVRPIKE, J) AVRPIKF, K) AVRPIZT, L) MAX60, M) MAX47, N) MAX67.

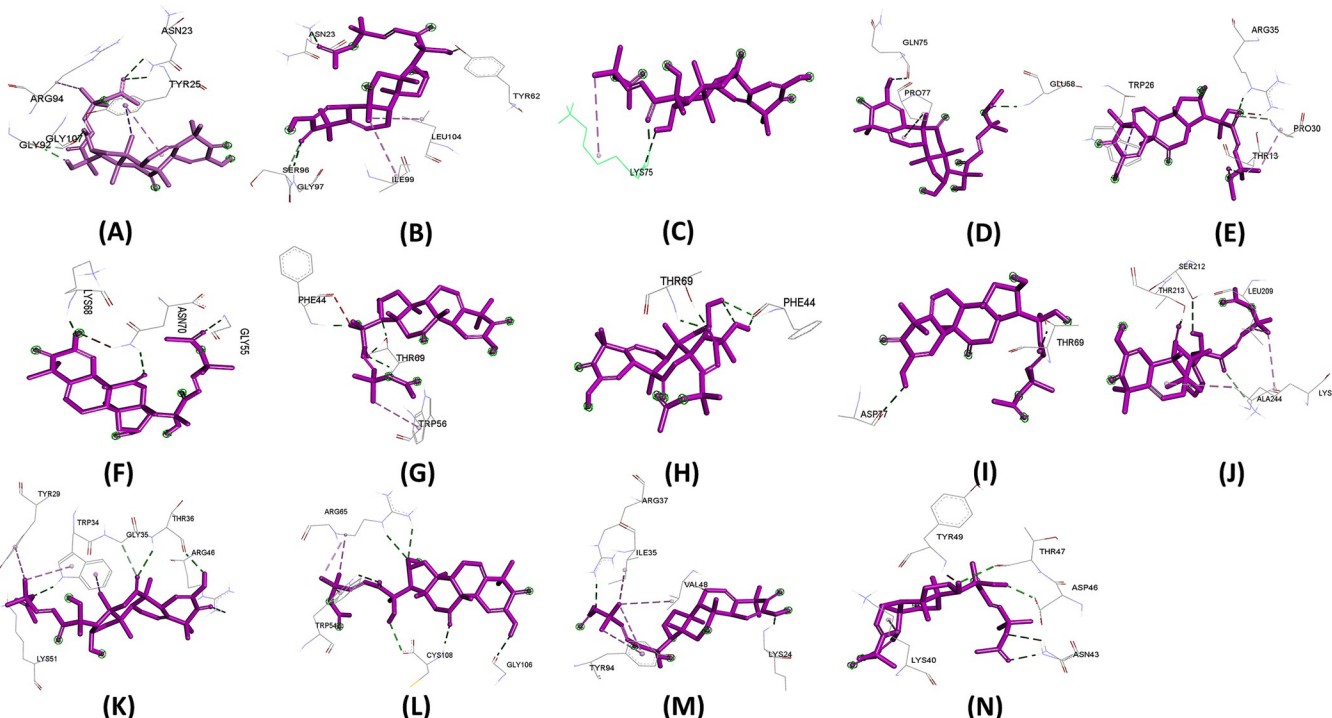

**Fig 4.** Binding site of ligand Cucurbitacin E with protein A) APIKL2A, B) APIKL2F, C) AVRPIA, D) AVRPIB, E) AVRPII, F) AVRPIKA, G) AVRPIKC, H) AVRPIKD, I) AVRPIKE, J) AVRPIKF, K) AVRPIZT, L) MAX60, M)MAX47, N) MAX67.

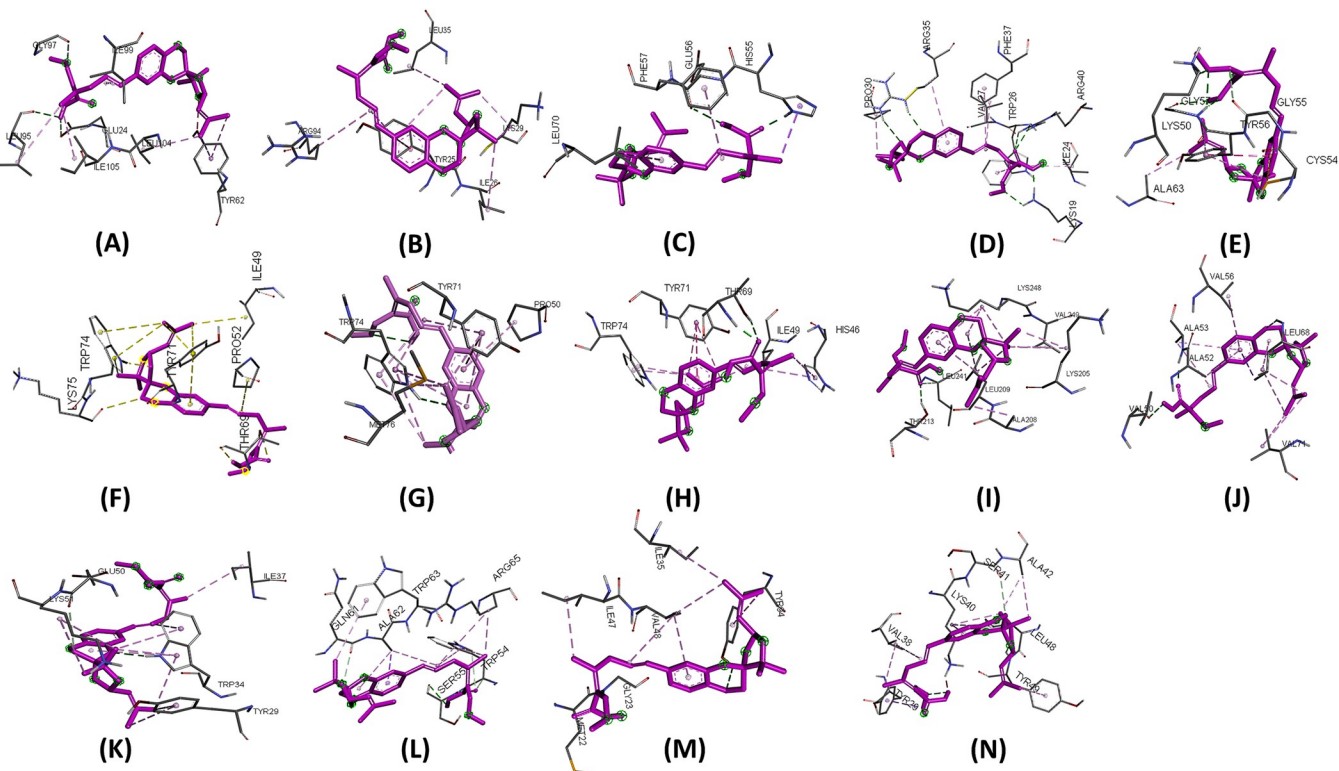

**Fig 5.** Binding site of ligand Strobilurin (control) with protein A) APIKL2A, B) APIKL2F, C) AVRPIA, D) AVRPIB, E) AVRPII, F) AVRPIKA, G) AVRPIKC, H) AVRPIKD, I) AVRPIKE, J) AVRPIKF, K) AVRPIZT, L) MAX60, M) MAX47, N) MAX67.

the case of AVR-Pia, the complex with HEC showed reasonably stable RMSD compared to the complex with STR, which showed significantly larger RMSD after around 45 ns until the end of the simulation. The RMSD in AVR-Pii was significantly larger for both the complexes with HEC and STR. The AVR-PikC complex with HEC showed significant deviations after around 40 ns until the end of the simulation compared to relatively stable and converged RMSD in the complex with STR. Both the complexes of AVR-PikD showed reasonably stable RMSD with an average of around 0.3 nm. The complexes of AVR-PikE and AVR-PikF with HEC and STR showed substantial deviations throughout the simulation period. However, the AVR-PikE complex with HEC showed slightly lower RMSD stabilizing after around 40 ns to an average of around 0.3 nm. AVR-PizT in complexes with HEC showed comparably lower RMSD than in complexes with STR. The complexes of MAX47 showed substantially more significant deviations than those with HEC and STR. In the case of MAX60, the RMSD in the complex with STR was slightly lower, with an average of around 0.4 nm, compared to the complex with HEC with substantially larger RMSD, reaching a maximum of around 0.9 nm.

The results of RMSD in ligand atoms showed that the ligands in complex with ApikL-2a, AVR-Pib, AVR-PikA, AVR-PizR, and MAX67 had significant deviations relative to the backbone atoms (Fig 7). In the case of ApikL-2f, the average RMSD for HEC and STR was around 1 nm. The RMSD was slightly higher and was around 2 nm for the ligands in complex with AVR-Pia and AVR-Pii. In the case of ACR-PikC compared to STR, the ligand HEC had significantly higher RMSD, where RMSD for STR was around 1 nm. In the case of AVR-PikE, both the ligands had lower and almost stable RMSD with an average of around 0.5 nm. The RMSD was relatively stable until around 80 ns in STR atoms in complex with MAX47, while in HEC,

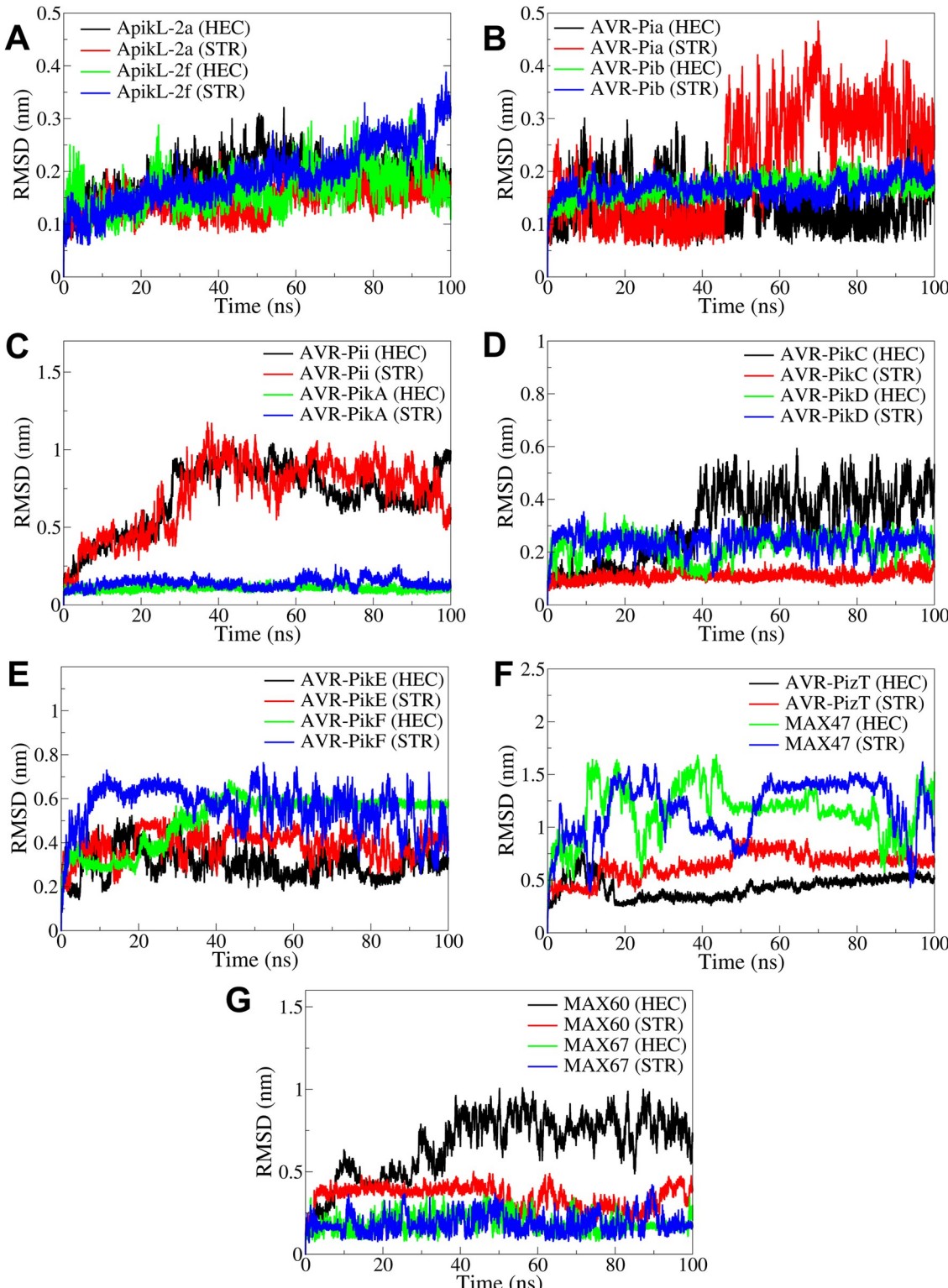

**Fig 6. The RMSD analysis in protein backbone atoms.** RMSD in backbone atoms of A) ApikL-2a and ApikL-2f, B) AVR-Pia and AVR-Pib, C) AVR-Pii and AVR-PikA, D) AVR-PikC and AVR-PikD, E) AVR-PikE and AVR-PikF, F) AVR-PizT and MAX47, and G) MAX60 and MAX67.

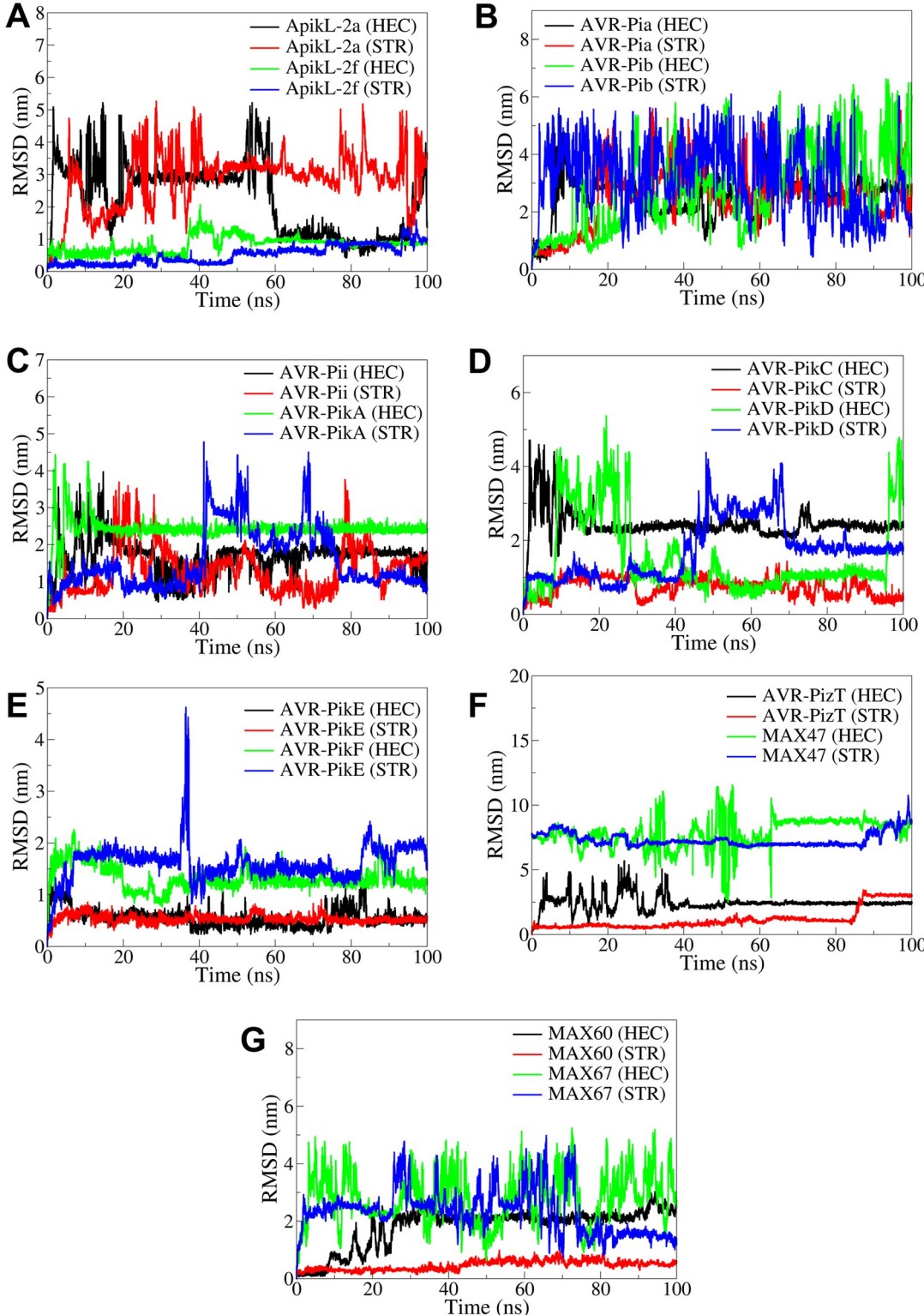

**Fig 7. The RMSD analysis in UNK atoms.** RMSD in UNK atoms in complexes of A) ApikL-2a and ApikL-2f, B) AVR-Pia and AVR-Pib, C) AVR-Pii and AVR-PikA, D) AVR-PikC and AVR-PikD, E) AVR-PikE and AVR-PikF, F) AVR-PizT and MAX47, and G) MAX60 and MAX67.

the RMSD stabilized to an average of around 0.2 nm after around 40 ns. The RMSD in STR atoms in the complex with MAX60 was relatively stable and lowest, with an average of around 0.5 nm, while the RMSD in HEC was significantly higher, with an average of around 2 nm.

**3.4.2. Root mean square fluctuation.** The RMSF analysis results suggested that for the RMSF in side chain atoms of respective proteins in the complex of both the ligands with ApikL-2a, ApikL-2f, AVR-PikA, AVR-PikC, AVR-PikD, AVR-PikE, AVR-PikF, AVR-PizT, MAX-60, and MAX67 were below 0.4 nm, excluding the higher RMSD in few of the terminal residues (Fig 8). The RMSF in the complexes with AVR-Pii and MAX47 was significantly higher. In the case of complexes with ApikL-2a and ApikL-2f, the residues in the range 25–50 showed significant RMSF, whereas the remaining residues showed stability below 0.2 nm RMSF. The complex of AVR-Pib with STR showed significantly higher RMSD in the residues in the range 40–50. The complexes with AVR-PikA, AVR-PikC, and AVR-PikD showed significant fluctuations in the residues in the range 25–50, while the fluctuation in the other residues, except terminal residues, was well below 0.2 nm. The RMSF was found considerably stable in AVR-PizT, MAX-60, and MAX67 complexes.

**3.4.3. Radius of gyration.** The radius of gyration results showed varied Rg in different complexes. The Rg in ApikL-2a and ApikL-2f were 1.3 to 1.4 nm (Fig 9). However, the ApikL-2f showed slight deviations after around 50 ns in both the complexes with HEC and STR. The Rg in AVR-Pia was almost stable for both the complexes with HEC and STR, while the Rg in AVR-Pib was lower and stable. The complexes of AVR-Pii showed significant deviations in Rg for both the complexes after around 20 ns simulation period. Meanwhile, the complexes of AVR-PikA showed stable Rg. In the case of AVR-PikC, the complex with HEC showed significantly higher Rg after around 40 ns, compared to the stable Rg in the complex with STR. Meanwhile, the Rg in both AVR-PikD complexes was stable. The Rg in AVR-PikE complexes was lower and stable throughout the simulation. Meanwhile, the Rg in AVR-PikF significantly deviated throughout the simulation for the complex with STR compared to the stable Rg in the complex with HEC. The Rg in AVR-PizT was lower and stable in both complexes, whereas the Rg in both complexes of MAX47 deviated significantly throughout the simulation. The Rg in the MAX60 complex with STR was stable compared to the complex with HEC, which showed significant deviations after around 40 ns simulation period. The Rg in both the complexes of MAX67 were lower and stable in both the complexes.

**3.4.4. Hydrogen bond analysis.** All the complexes investigated showed at least one hydrogen bond consistently formed throughout the simulation period (Fig 10). A maximum of two frequent hydrogen bonds were observed in the complexes of ApikL-2a and APikL-2f, where STR showed more hydrogen bonds for a brief period of 25 to 35 ns in the case of APikL-2f. Compared to the complex of AVR-Pia with STR, the complex with HEC showed more number and more frequent hydrogen bonds. Similarly, in the AVR-Pib complex with STR, HEC showed more and more frequent hydrogen bonds. In the case of AVR-Pii complexes, HEC showed more hydrogen bonds than STR, while in the case of complexes with AVR-PikA, STR showed more frequent hydrogen bonds. In the case of complexes of AVR-PikC and AVR-PikD, both the ligands formed a maximum of two hydrogen bonds frequently, except the STR in AVR-PikD, which showed a number of hydrogen bonds during the first 40 ns simulation period. Both the ligands formed two hydrogen bonds frequently in the complexes of AVR-PikE and AVR-PikF. In the case of AVR-PizT, both the ligands formed a maximum of two hydrogen bonds quite frequently throughout the simulation. The ligand STR formed more consistent and frequent hydrogen bonds with MAX47 than HEC. The ligands formed a maximum of three hydrogen bonds during the first 25 ns simulation period in MAX60 and MAX67 complexes. However, after 25 ns, the ligand STR showed consistent hydrogen bonds compared to HEC in these complexes.

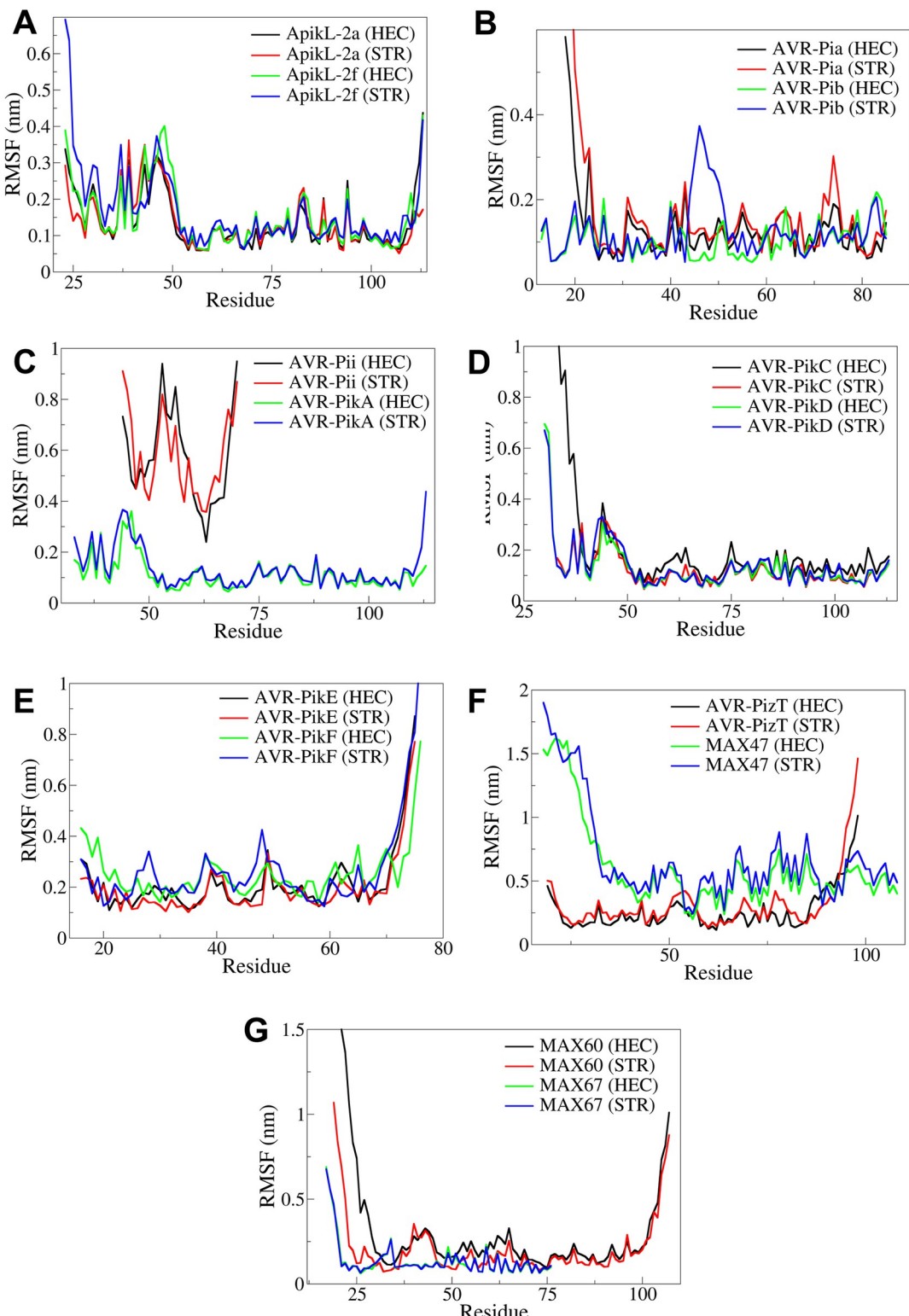

**Fig 8.** The RMSF analysis in the complexes of A) ApikL-2a and ApikL-2f, B) AVR-Pia and AVR-Pib, C) AVR-Pii and AVR-PikA, D) AVR-PikC and AVR-PikD, E) AVR-PikE and AVR-PikF, F) AVR-PizT and MAX47, and G) MAX60 and MAX67.

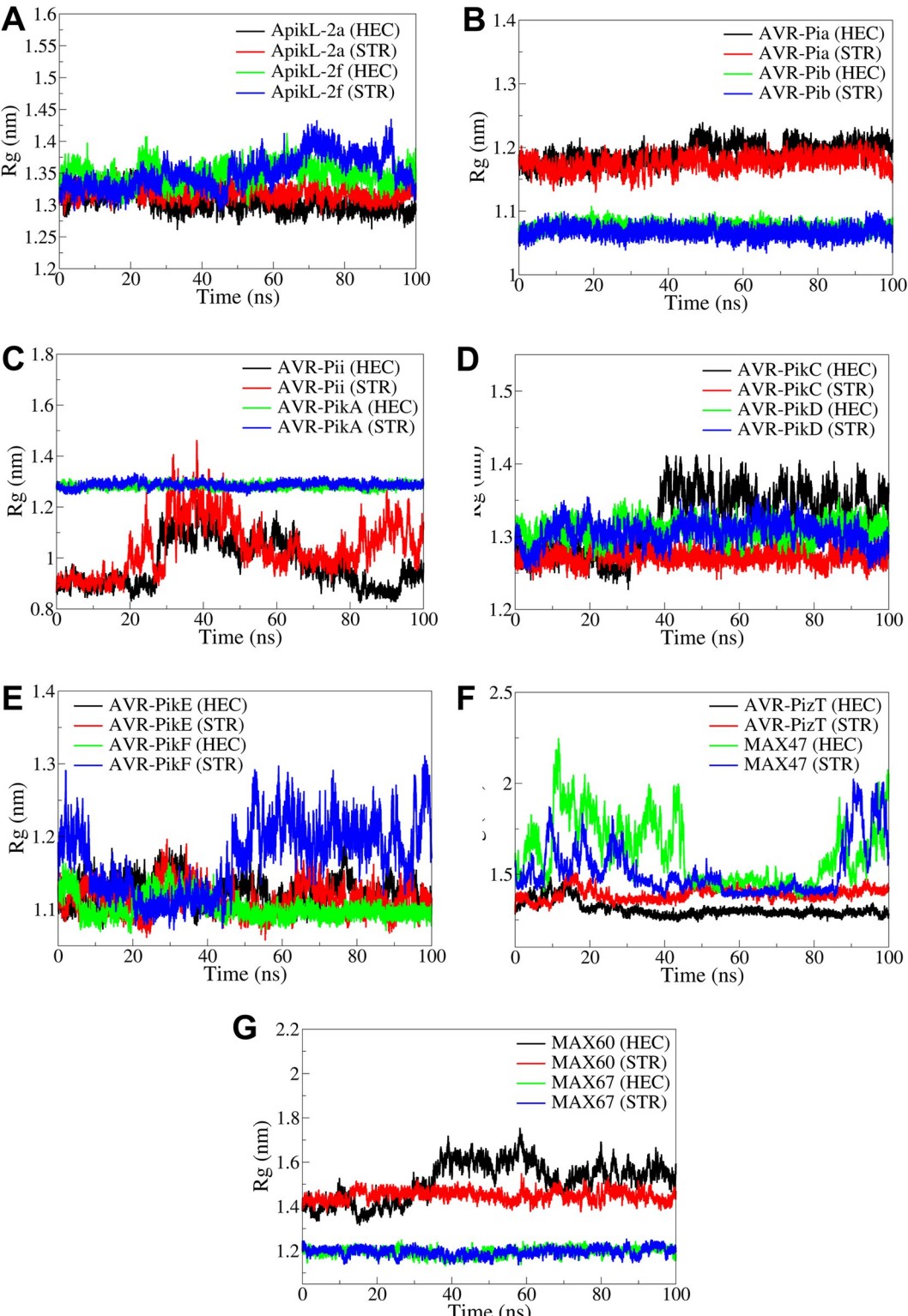

**Fig 9.** The Rg analysis in the complexes of A) ApikL-2a and ApikL-2f, B) AVR-Pia and AVR-Pib, C) AVR-Pii and AVR-PikA, D) AVR-PikC and AVR-PikD, E) AVR-PikE and AVR-PikF, F) AVR-PizT and MAX47, and G) MAX60 and MAX67.

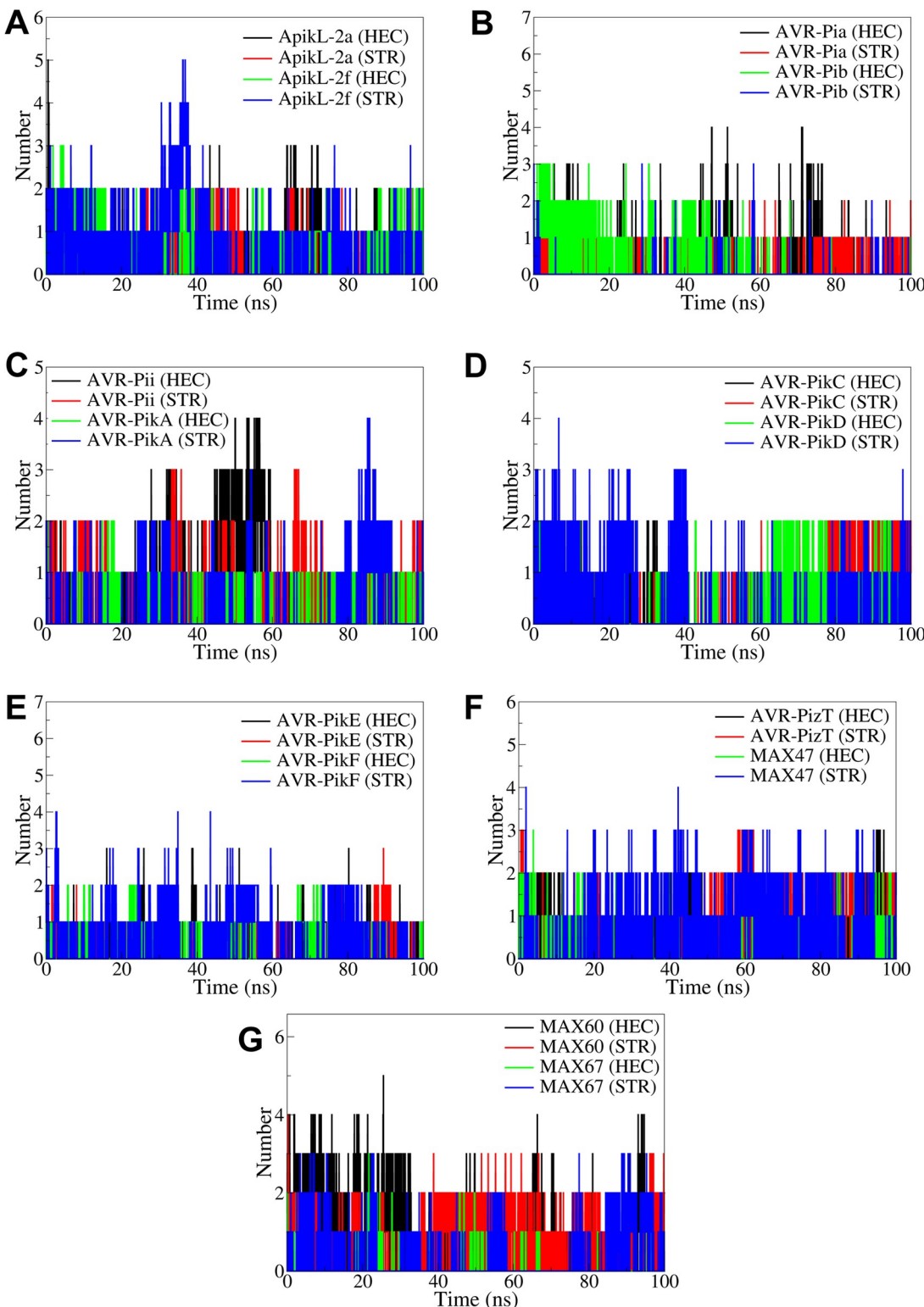

**Fig 10.** Hydrogen bond analysis in the complexes of A) ApikL-2a and ApikL-2f, B) AVR-Pia and AVR-Pib, C) AVR-Pii and AVR-PikA, D) AVR-PikC and AVR-PikD, E) AVR-PikE and AVR-PikF, F) AVR-PizT and MAX47, and G) MAX60 and MAX67.

In the ApikL-2a complex, the ligand HEC formed hydrogen bonds with Lys109 and Gly92 at the simulation time intervals 0 and 25 ns (S3 Fig). However, the 50, 75, and 100 trajectories showed no hydrogen bonds. The ligand STR showed the hydrogen bonds with Gly97 and Leu95 were shown only in 0 and 25 ns trajectories. In the case of the ApikL-2f complex, the ligand HEC formed hydrogen bonds with Leu95 in the 0 and 25 ns trajectories, while with Met108 in the 75 ns trajectory, and Arg94 and Leu35 in the 100 ns trajectory. The complex with STR showed a hydrogen bond with Asn23 in 0 and 25 ns trajectories, while the 75 ns trajectory showed a hydrogen bond with Glu24. AVR-Pia only equilibrated from showed a hydrogen bond between HEC and Asn72 (S4 Fig). At the same time, STR formed hydrogen bonds with Glu58, Glu56, and His55 in an equilibrated trajectory, with residue Thr47 at 25 ns and residue Agr43 at 75 ns. In the case of AVR-Pib, only the equilibrated trajectory showed hydrogen bonds between HEC and Lys60, while the STR showed hydrogen bonds with Lys19 in equilibrated and 25 ns trajectory. In the complex of AVR-Pii, the ligand HEC showed no hydrogen bonds with Asp59, Cys51, and Cys69 in the 50 ns trajectory, while the trajectories at other time steps showed no hydrogen bonds (S5 Fig). The ligand STR showed hydrogen bonds with Gly57, Thr56, and Lys68 in the equilibrated trajectory and Trp74 and Thr69 in the 25 ns trajectory. The 50 and 75 ns trajectories showed hydrogen bonds between STR and Ser52 and Asn70, respectively. The 100 ns trajectory showed hydrogen bonds with Tyr64 and Asp62. In the case of the complex with AVR-PikA, the ligand HEC showed no hydrogen bonds in the extracted trajectories. The ligand STR showed hydrogen bonds with Trp74 and Thr69 at 0 ns and 25 ns and residues Asn46 and Asp45 at 100 ns.

In the AVR-PikC complex, the ligand HEC formed hydrogen bonds with Lys109 in equilibrated and 25 ns trajectory and Thr69 in 100 ns trajectory (S6 Fig). Meanwhile, the ligand STR only showed a hydrogen bond with Asn46 in the 75 ns trajectory. In the case of the AVR-PikD complex, the ligand HEC formed hydrogen bonds with Lys75 and Phe44 in the equilibrated and 25 ns trajectory and with Thr69 in the 75 ns trajectory. The ligand STR showed hydrogen bonds with Trp74, Thr69, and Asn42 residues in equilibrated and 25 ns trajectories and Asn83 in 50 ns trajectory. In the case of AVR-PikE, only the ligand HEC showed a hydrogen bond with Gly235 in an equilibrated and 25 ns trajectory (S7 Fig). In the case of the complex of AVR-PikF, the ligand HEC formed a hydrogen bond with Ser18 in 25 and 50 ns trajectories. Meanwhile, the ligand STR formed hydrogen bonds with Val50, Ala52, and Leu68 in equilibrated and 25 ns trajectories and with Asn51 residue in 75 ns trajectory.

In the case of AVR-PizT complexes, the ligand HEC formed a hydrogen bond with residue His33 in equilibrated and 25 ns trajectories and with Leu58 in 75 and 100 ns trajectories (S8 Fig). Meanwhile, the ligand STR formed a hydrogen bond with Arg46 in equilibrated and 25 ns trajectories and Val96 and Val21 residues in 100 ns trajectory. The equilibrated and 25 ns trajectories of the MAX47 complex with HEC showed hydrogen bonds with Trp63 and Trp54. The ligand STR showed a hydrogen bond with Ile53 in equilibrated and 25 ns trajectories, Trp63 in 50 ns trajectory, and Lys57 in 75 ns trajectory. In the case of complexes of MAX60, the ligand HEC formed a hydrogen bond with Asp56 in equilibrated and 25 ns trajectories, with Asn101 in 50 ns trajectory, and with Asn101 and Gln97 residues in 100 ns trajectory (S9 Fig). The ligand STR formed hydrogen bonds with Tyr94 and Gly23 in equilibrated and 25 ns trajectories, with Gln97 in 50 ns trajectory and Gln45 in 100 ns trajectory. In the complex of MAX67, the ligand HEC formed a hydrogen bond with Thr47 in equilibrated and 25 ns trajectory, while STR formed a hydrogen bond with Lys40 in equilibrated and 25 ns trajectory.

**3.4.5. MM-GBSA calculations.** The results of MM-GBSA calculations are given in Table 2. The ligand HEC showed comparably more favorable binding affinities compared to STR in terms of lower $\Delta G_{binding}$ in the complexes of ApikL-2a, AVR-Pia, AVR-Pii, AVR-PikA,

**Table 2. MM-GBSA results.**

| Complex | Energy component (kcal/mol) | | | | | | | | |
|---|---|---|---|---|---|---|---|---|---|
| | ENTROPY (-TΔS) | ΔVDWAALS | ΔEEL | ΔEGB | ΔESURF | ΔGGAS | ΔGSOLV | ΔTOTAL | ΔG binding |
| ApikL-2a (HEC) | 7.03 (0.04) | -12.39 (0.71) | -0.41 (0.36) | 2.28 (0.06) | -1.73 (0.48) | -12.80 (1.33) | 0.55 (0.49) | -12.25 (1.42) | -1.17 (6.83) |
| ApikL-2a (STR) | 7.61 (0.05) | -10.80 (0.29) | -1.10 (0.44) | 2.46 (0.08) | -1.56 (0.21) | -11.91 (0.91) | 0.90 (0.22) | -11.01 (0.93) | 3.65 (7.06) |
| ApikL-2f (HEC) | 4.64 (0.05) | -24.86 (1.09) | -0.98 (0.18) | 4.20 (0.07) | -3.28 (0.05) | -25.84 (1.56) | 0.93 (0.09) | -24.91 (1.57) | -16.21 (4.44) |
| ApikL-2f (STR) | 8.01 (6.66) | -32.45 (1.18) | -1.43 (0.44) | 5.32 (0.13) | -4.81 (0.04) | -33.87 (1.46) | 0.51 (0.14) | -33.36 (1.47) | -17.87(10.67) |
| AVR-Pia (HEC) | 5.58 (0.05) | -11.20 (0.60) | -0.40 (0.37) | 2.05 (0.07) | -1.51 (0.29) | -11.60 (1.27) | 0.55 (0.30) | -11.06 (1.30) | -3.87 (5.44) |
| AVR-Pia (STR) | 8.74 (0.05) | -11.17 (0.32) | -1.46 (0.13) | 2.62 (0.45) | -1.61 (0.60) | -12.62 (0.81) | 1.01 (0.75) | -11.61 (1.10) | 2.34 (8.01) |
| AVR-Pib (HEC) | 6.32 (0.05) | -2.98 (0.23) | -0.29 (0.09) | 0.94 (0.46) | -0.38 (0.28) | -3.26 (1.16) | 0.56 (0.54) | -2.70 (1.28) | 6.86 (5.22) |
| AVR-Pib (STR) | 7.36 (0.07) | -4.04 (1.23) | -0.37 (0.56) | 0.96 (0.25) | -0.56 (0.20) | 4.41 (1.55) | 0.40 (0.31) | -4.01 (1.59) | -3.57 (6.94) |
| AVR-Pii (HEC) | 5.59 (0.05) | -20.23 (1.44) | -1.67 (0.18) | 3.98 (0.09) | -2.69 (0.01) | -21.90 (1.73) | 1.29 (0.09) | -20.60 (1.73) | -3.81 (5.27) |
| AVR-Pii (STR) | 8.72 (0.05) | -19.59 (0.69) | -1.24 (0.30) | 3.40 (0.16) | -2.84 (0.21) | -20.82 (1.02) | 0.56 (0.26) | -20.26 (1.05) | -1.19 (8.38) |
| AVR-PikA (HEC) | 2.73 (0.04) | -20.19 (1.40) | -1.10 (0.30) | 3.56 (0.06) | -2.35 (0.01) | -21.29 (1.79) | 1.21 (0.06) | -20.08 (1.79) | -14.12 (2.53) |
| AVR-PikA (STR) | 9.49 (0.05) | -16.38 (0.87) | -1.23 (0.48) | 2.97 (0.24) | -2.26 (0.25) | -17.61 (1.23) | 0.71 (0.34) | -16.90 (1.27) | -5.73 (9.15) |
| AVR-PikC (HEC) | 5.89 (0.05) | -22.56 (0.37) | -1.81 (0.13) | 4.85 (0.41) | -2.94 (0.36) | -24.37 (1.14) | 1.91 (0.55) | -22.46 (1.26) | -9.73 (5.39) |
| AVR-PikC (STR) | 4.67 (0.04) | -25.69 (0.59) | -0.73 (0.38) | 3.48 (0.06) | -3.75 (0.23) | -26.42 (1.02) | -0.27(0.24) | -26.69 (1.04) | -9.14 (4.72) |
| AVR-PikD (HEC) | 5.88 (5.36) | -17.99 (0.30) | -1.01 (0.29) | 3.39 (0.12) | -2.49 (0.27) | -18.99 (1.16) | 0.91(0.30) | -18.09 (1.20) | -8.80 (7.76) |
| AVR-PikD (STR) | 7.82 (0.05) | -17.98 (0.84) | -4.28 (0.58) | 6.12 (0.07) | -2.29 (0.25) | -22.26 (1.27) | 3.82 (0.26) | -18.43 (1.29) | -3.00 (6.13) |
| AVR-PikE (HEC) | 3.20 (0.05) | -18.94 (0.66) | -1.11(0.02) | 3.46 (0.37) | -2.71 (0.08) | -20.05 (1.22) | 0.75 (0.38) | 19.31 (1.27) | -9.79 (2.70) |
| AVR-PikE (STR) | 3.87 (0.05) | -31.50(0.66) | -2.63(0.46) | 5.02 (0.17) | -4.58 (0.02) | -34.13 (1.08) | 0.44 (0.17) | -33.69 (1.09) | -21.97 (3.61) |
| AVR-PikF (HEC) | 2.88 (0.05) | -23.21 (1.09) | -2.57 (0.34) | 5.34 (0.14) | -3.17 (0.02) | -25.78 (1.53) | 2.18 (0.15) | -23.60 (1.54) | -16.12 (2.66) |
| AVR-PikF (STR) | 6.25 (1.79) | -22.46 (0.86) | -1.66 (0.39) | 3.63 (0.05) | -3.17 (0.05) | -24.13 (1.20) | 0.47 (0.07) | -23.66 (1.21) | -7.45 (6.34) |
| AVR-PizT (HEC) | 2.99 (0.05) | -19.31 (1.43) | -1.49 (0.36) | 3.50 (0.07) | -2.80 (0.02) | -20.81 (1.80) | 0.70 (0.07) | -20.11 (1.80) | -13.18 (3.03) |
| AVR-PizT (STR) | 7.20 (0.05) | -22.73 (1.64) | -1.47 (0.02) | 4.03 (0.46) | -3.31 (0.01) | -24.20 (1.80) | 0.72 (0.46) | -23.49 (1.86) | -5.72 (7.04) |
| MAX47 (HEC) | 6.60 (0.24) | -11.70 (1.32) | -0.90 (0.38) | 2.40 (0.10) | -1.49 (0.14) | 12.60 (1.75) | 0.90 (0.17) | -11.70 (1.76) | -4.98 (6.04) |
| MAX47 (STR) | 17.57 (0.56) | -34.95 (9.57) | -4.31(0.93) | 8.23 (2.66) | -5.02 (1.74) | -39.27 (9.64) | 3.21(3.18) | -36.06(10.15) | 3.24 (16.14) |
| MAX60 (HEC) | 5.17 (5.50) | -24.26 (1.13) | -1.33 (0.19) | 4.43 (0.07) | -3.60 (0.08) | -25.59 (1.58) | 0.83 (0.11) | -24.76 (1.59) | -11.00 (7.62) |
| MAX60 (STR) | 4.87 (0.05) | -30.77 (0.90) | -1.67 (0.11) | 5.09 (0.24) | -4.63 (0.13) | -32.44 (1.16) | 0.46 (0.28) | -31.98(1.19) | -19.80 (4.39) |
| MAX67 (HEC) | 4.05 (0.05) | -2.57 (0.58) | -0.08 (0.27) | 0.54 (0.03) | -0.35 (0.10) | -2.65 (1.18) | 0.19 (0.10) | -2.46 (1.19) | 3.66 (3.86) |
| MAX67 (STR) | 11.42 (0.05) | -12.61 (1.21) | -0.60 (0.38) | 1.92 (0.03) | -1.70 (0.18) | -13.21 (1.46) | 0.22 (0.18) | -12.99 (1.48) | -6.06 (11.63) |

ΔVDWAALS: van der Waals energy; ΔEEL: Electrostatic energies; ΔEGB: Polar solvation energy; ΔESURF: Nonpolar solvation energy; ΔGGAS = ΔVDWAALS+ΔEEL; ΔGSOLV = ΔEGB+ ΔESURF; ΔTOTAL = ΔGSOLV +ΔGGAS; ΔG binding = ΔTOTAL–TΔS (Standard deviations are given in parentheses)

AVR-PikC, AVR-PikD, AVR-PikF, AVR-PizT, and MAX47. In all these complexes, either due to favorable entropic energies, due to lower van der Waals energies, or due to higher electrostatic energies, the relative binding energies (ΔTOTAL) and the binding free energy (ΔG$_{binding}$) were lower for HEC, notably, in the case of ApikL-2a, AVR-Pia, and MAX47, the ligand HEC had comparably and significantly lower ΔG$_{binding}$. However, in the complexes with AVR-Pib, AVR-PiE, MAX60, and MAX67, HEC had higher ΔG$_{binding}$ than the STR. The most favorable ΔG$_{binding}$ for the ligand HEC was observed with AVR-PikA, AVR-PikF, and AVR-PizT with ΔG$_{binding}$ of -14.12, -16.12, and -13.18 kcal/mol, respectively.

## 3.5. Phytochemical properties

The phytochemical properties of the compounds were analyzed using the SwissADME server. Among the 35 metabolites, a few showed negative phytochemical properties and were thus excluded from consideration as top metabolites. Hecogenin and Cucurbitacin E were further evaluated using Lipinski's Rule of 5. Table 3 demonstrates that these target molecules can be

**Table 3. Physiochemical properties of top metabolites.**

| Properties | Hecogenin | Cucurbitacin E |
|---|---|---|
| Formula | C27H42O4 | C32H44O8 |
| MW | 430.62 | 556.69 |
| Heavy atoms | 31 | 40 |
| Aromatic heavy atoms | 0 | 0 |
| Fraction Csp3 | 0.96 | 0.69 |
| Rotatable bonds | 0 | 6 |
| H-bond acceptors | 4 | 8 |
| H-bond donors | 1 | 3 |
| MR | 122.27 | 150.88 |
| TPSA | 55.76 | 138.2 |
| iLOGP | 4.06 | 3.7 |
| XLOGP3 | 4.83 | 3.25 |
| WLOGP | 4.97 | 4.19 |
| MLOGP | 4.09 | 1.68 |
| Silicos-IT Log P | 3.99 | 4.49 |
| Consensus Log P | 4.39 | 3.46 |
| ESOL Log S | -5.55 | -4.94 |
| ESOL Solubility (mg/ml) | 1.21E-03 | 6.35E-03 |
| ESOL Solubility (mol/l) | 2.80E-06 | 1.14E-05 |
| ESOL Class | Moderately soluble | Moderately soluble |
| Ali Log S | -5.73 | -5.83 |
| Ali Solubility (mg/ml) | 7.94E-04 | 8.31E-04 |
| Ali Solubility (mol/l) | 1.84E-06 | 1.49E-06 |
| Ali Class | Moderately soluble | Moderately soluble |
| Silicos-IT LogSw | -4.38 | -4.25 |
| Silicos-IT Solubility (mg/ml) | 1.78E-02 | 3.15E-02 |
| Silicos-IT Solubility (mol/l) | 4.13E-05 | 5.66E-05 |
| Silicos-IT class | Moderately soluble | Moderately soluble |
| Lipinski violation | 0 | 1 |
| Ghose violation | 1 | 3 |
| Veber violation | 0 | 0 |
| Egan violation | 0 | 1 |
| Muegge violation | 0 | 0 |
| Bioavailability Score | 0.55 | 0.55 |
| PAINS alerts | 0 | 0 |
| Brenk | 0 | 2 |
| Leadlikeness violations | 2 | 1 |
| Synthetic Accessibility | 6.7 | 6.74 |

considered safe for treatment. Their molecular weights range between 430 and 557 Da, and both ligands exhibit acceptable solubility. Additionally, they show good H-bond acceptor and donor characteristics, with a bioavailability score of less than 0.6.

## 3.6. Toxicity analysis

The toxicity analysis of both top metabolites was conducted using the Deep-PK prediction server. The compounds were evaluated for potential AMES toxicity, carcinogenicity, and any irritation to the environment or human health (Table 4). The AMES mutagenesis probability

**Table 4. Toxicity analysis of top metabolites.**

| | Hecogenin | Cucurbitacin E |
|---|---|---|
| [Toxicity/AMES Mutagenesis] Predictions | Safe | Safe |
| [Toxicity/AMES Mutagenesis] Probability | 0.126 | 0.009 |
| [Toxicity/AMES Mutagenesis] Interpretation | Safe (High Confidence) | Safe (High Confidence) |
| [Toxicity/Avian] Predictions | Safe | Safe |
| [Toxicity/Avian] Probability | 0.061 | 0.011 |
| [Toxicity/Avian] Interpretation | Safe (High Confidence) | Safe (High Confidence) |
| [Toxicity/Biodegradation] Predictions | Safe | Safe |
| [Toxicity/Biodegradation] Probability | 0.016 | 0.002 |
| [Toxicity/Biodegradation] Interpretation | Safe (High Confidence) | Safe (High Confidence) |
| [Toxicity/Carcinogenesis] Predictions | Safe | Toxic |
| [Toxicity/Carcinogenesis] Probability | 0.169 | 0.639 |
| [Toxicity/Carcinogenesis] Interpretation | Safe (Medium Confidence) | **Toxic** (Low Confidence) |
| [Toxicity/Liver Injury I] Predictions | Safe | Safe |
| [Toxicity/Liver Injury I] Probability | 0.474 | 0.385 |
| [Toxicity/Liver Injury I] Interpretation | Safe (Low Confidence) | Safe (Low Confidence) |
| [Toxicity/Eye Corrosion] Predictions | Safe | Safe |
| [Toxicity/Eye Corrosion] Probability | 0 | 0 |
| [Toxicity/Eye Corrosion] Interpretation | Safe (High Confidence) | Safe (High Confidence) |
| [Toxicity/Eye irritation] Predictions | Safe | Safe |
| [Toxicity/Eye irritation] Probability | 0.022 | 0.003 |
| [Toxicity/Eye irritation] Interpretation | Safe (High Confidence) | Safe (High Confidence) |
| [Toxicity/hERG Blockers] Predictions | Safe | Safe |
| [Toxicity/hERG Blockers] Probability | 0.023 | 0.001 |
| [Toxicity/hERG Blockers] Interpretation | Safe (High Confidence) | Safe (High Confidence) |
| [Toxicity/Daphnia Maga] Predictions | 5.14 | 6.93 |
| [Toxicity/Daphnia Maga] Interpretation | None | None |

ranged from 0.1 to 0.009, which is considered safe for the environment. The biodegradation probability ranged from 0.061 to 0.011. Additionally, the metabolites were predicted to be safe in terms of avian toxicity and carcinogenicity.

## 4. Discussion

Rice blast is a major agricultural disease causing significant financial losses worldwide. Genetic and genomic investigations have improved our understanding of the disease, with numerous genes implicated in its pathogenesis and development. Various virulence factors and signaling pathways have been identified, suggesting that *M. oryzae* employs multiple strategies to infiltrate the host and weaken its defense mechanisms.

Effector proteins, including AvrPik variants and MAX proteins, have been targeted in this study due to their ability to subvert rice defenses by interacting with Pik alleles on chromosome 11. The Pik locus includes genes such as Pik*, Pikm, and Pikp, which recognize AvrPik variants A to F with slight amino acid differences [45]. The interaction, mediated by the HMA domain of Pik-1, has been extensively studied, especially for its role in host-pathogen interactions [46–48]. Advances in engineering receptor specificities have been promising for developing resistance against various pathogen strains.

In this study, we identified 35 compounds with potential inhibitory effects against *M. oryzae*, with Hecogenin and Cucurbitacin E emerging as the most promising candidates based on

molecular docking. These compounds demonstrated strong binding affinity with key proteins such as MAX40 and APIKL2A, which are critical in fungal transcription and host immunity suppression. Importantly, both compounds exhibited higher binding affinity than the reference metabolite, Strobilurin, suggesting their potential as effective fungicidal agents. Additionally, SwissADME server analysis confirmed favorable phytochemical properties of both compounds, highlighting their suitability for fungicide development.

The selection of Hecogenin and Cucurbitacin E is supported by existing studies, which have reported their antifungal and anti-inflammatory properties, aligning with our findings of high binding affinities to rice blast effector proteins. For instance, Hecogenin has shown promising antifungal activities, making it a valuable lead in agricultural applications. Similarly, Cucurbitacin E is known for its bioactive properties, including potential fungicidal effects, suggesting that its mechanism of action may also extend to inhibiting fungal effectors [49]. However, as no specific experimental studies have yet assessed these compounds against *M. oryzae*, our computational approach offers novel insights that warrant future in vitro or in vivo investigations.

To validate the computational findings, MD simulations provided insights into the stability and binding dynamics of these compounds with effector proteins. RMSD analysis indicated that complexes involving AVR-Pikb, AVR-PikA, and AVR-PikC with either Hecogenin or Strobilurin showed high stability, with RMSD values below 0.2 nm, supporting the strong binding observed in docking studies [50]. Additionally, RMSF analysis highlighted that loop and terminal regions contributed to the flexibility, while the core residues remained stable in complexes, affirming the strong binding of both ligands with specific protein residues [51].

The MM-GBSA calculations, used to estimate binding free energies, further substantiated the favorable binding of Hecogenin and Cucurbitacin E compared to Strobilurin in proteins such as ApikL-2a and AVR-PikC. This approach, which integrates van der Waals, electrostatic, and solvation energies, has been validated in previous studies as a reliable method for assessing ligand binding affinity [52].

Our findings, aligned with Lipinski's rule of five, indicate that these natural compounds possess suitable pharmacokinetic properties for fungicide development. Both compounds' molecular weights, LogP values, and hydrogen bond acceptors and donors fall within acceptable ranges, making them favorable candidates for plant-based fungicides [53]. Moreover, the toxicity analysis using the Deep-PK prediction server revealed that the mutagenesis and avian toxicity probabilities are low, supporting the safe application of these compounds as potential fungicides in rice blast management.

Consequently, while existing literature supports the antifungal potential of Hecogenin and Cucurbitacin E, this study offers novel insights into their specific interactions with *M. oryzae* effectors, justifying further experimental studies. Our computational approach lays the groundwork for their development as lead compounds in new fungicide formulations targeting rice blast disease, thus filling a critical research gap in combating this pathogen through innovative, plant-derived compounds.

## 5. Conclusion

This study identified Hecogenin and Cucurbitacin E as promising antifungal agents against *M. oryzae* based on extensive computational analyses, including molecular docking and molecular dynamics simulations. Both compounds displayed strong, consistent binding affinities with 14 key effector proteins, surpassing the binding strength of the reference compound, Strobilurin. These results suggest that Hecogenin and Cucurbitacin E have potential as effective fungicidal candidates for managing rice blast disease. Further in vitro and in vivo studies are needed to

validate these computational findings and assess the practical application of these compounds in agricultural settings. Additionally, optimizing their structural and functional properties could enhance their efficacy and safety for agricultural use.

## Supporting information

**S1 Table. Anti-viral and insecticidal Activity plant metabolites.**
(DOCX)

**S2 Table. Result of molecular docking of effector proteins and ligands.**
(DOCX)

**S1 Fig.** All significant 2D interaction of ligand Hecogenin with protein A) APIKL2A, B) APIKL2F, C) AVRPIA, D) AVRPIB, E) AVRPII, F) AVRPIKA, G) AVRPIKC, H) AVRPIKD, I) AVRPIKE, J) AVRPIKF, K) AVRPIZT, L) MAX60, M) MAX47, N) MAX67.
(DOCX)

**S2 Fig.** All 2D interaction of ligand Cucurbitacin E with protein A) APIKL2A, B) APIKL2F, C) AVRPIA, D) AVRPIB, E) AVRPII, F) AVRPIKA, G) AVRPIKC, H) AVRPIKD, I) AVR-PIKE, J) AVRPIKF, K) AVRPIZT, L) MAX60, M) MAX47, N) MAX67.
(DOCX)

**S3 Fig. Hydrogen bond interactions in trajectories at different time intervals.** A) ApikL-2a (HEC), B) Apikl-2a (STR), C) ApikL-2f (HEC), and D) Apikl-2f (STR).
(DOCX)

**S4 Fig. Hydrogen bond interactions in trajectories at different time intervals.** A) AVR-Pia (HEC), B) AVR-Pia (STR), C) AVR-Pib (HEC), and D) AVR-Pib (STR).
(DOCX)

**S5 Fig. Hydrogen bond interactions in trajectories at different time intervals.** A) AVR-Pii (HEC), B) AVR-Pii (STR), C) AVR-PikA, and D) AVR-PikA (STR).
(DOCX)

**S6 Fig. Hydrogen bond interactions in trajectories at different time intervals.** A) AVR-PikC (HEC), B) AVR-PikC (STR), C) AVR-PikD (HEC), and D) AVR-PikD (STR).
(DOCX)

**S7 Fig. Hydrogen bond interactions in trajectories at different time intervals.** A) AVR-PikE (HEC), B) AVR-PikE (STR), C) AVR-PikF (HEC), and D) AVR-PikF (STR).
(DOCX)

**S8 Fig. Hydrogen bond interactions in trajectories at different time intervals.** A) AVR-PizT (HEC), B) AVR-PizT (STR), C) MAX47 (HEC), and D) MAX47 (STR).
(DOCX)

**S9 Fig. Hydrogen bond interactions in trajectories at different time intervals.** A) MAX60 (HEC), B) MAX60 (STR), C) MAX67 (HEC), and D) MAX67 (STR).
(DOCX)

## Author Contributions

**Conceptualization:** Abu Tayab Moin.

**Formal analysis:** Tanjin Barketullah Robin, Abu Tayab Moin.

**Funding acquisition:** Jannatul Ferdousy, Mst Sanjida Nasrin, Istiak Ahmed.

**Methodology:** Tanjin Barketullah Robin, Nadim Ahmed, Nurul Amin Rani, Md Shishir Bhuyian, Suvarna N. Vakare, Abu Tayab Moin, Rajesh B. Patil.

**Project administration:** Tanjin Barketullah Robin, Abu Tayab Moin, Mohammad Shahadat Hossain.

**Resources:** Jannatul Ferdousy, Mst Sanjida Nasrin, Istiak Ahmed, Tawsif Hossain, Md Mehedi Hasan, Mehrab Hassan Soaeb, Md. Ahsanul Tamim, Nusrat Jahan Yeasmin, Ummay Habiba, Nadim Ahmed, Md Shishir Bhuyian, Suvarna N. Vakare, Abu Tayab Moin, Rajesh B. Patil.

**Software:** Suvarna N. Vakare, Rajesh B. Patil.

**Supervision:** Rajesh B. Patil, Mohammad Shahadat Hossain.

**Validation:** Jannatul Ferdousy, Mst Sanjida Nasrin, Istiak Ahmed, Tawsif Hossain, Nurul Amin Rani, Abu Tayab Moin, Rajesh B. Patil, Mohammad Shahadat Hossain.

**Visualization:** Md Shishir Bhuyian, Suvarna N. Vakare, Abu Tayab Moin, Rajesh B. Patil.

**Writing – original draft:** Jannatul Ferdousy, Tanjin Barketullah Robin, Mst Sanjida Nasrin, Istiak Ahmed, Tawsif Hossain, Md Mehedi Hasan, Mehrab Hassan Soaeb, Md. Ahsanul Tamim, Nusrat Jahan Yeasmin, Ummay Habiba, Nadim Ahmed, Nurul Amin Rani, Md Shishir Bhuyian, Suvarna N. Vakare, Abu Tayab Moin, Rajesh B. Patil, Mohammad Shahadat Hossain.

**Writing – review & editing:** Jannatul Ferdousy, Tanjin Barketullah Robin, Mst Sanjida Nasrin, Istiak Ahmed, Tawsif Hossain, Md Mehedi Hasan, Mehrab Hassan Soaeb, Md. Ahsanul Tamim, Nusrat Jahan Yeasmin, Ummay Habiba, Nadim Ahmed, Nurul Amin Rani, Md Shishir Bhuyian, Suvarna N. Vakare, Abu Tayab Moin, Rajesh B. Patil, Mohammad Shahadat Hossain.

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
