## [Decision Letter · Decision Letter 0]

29 Sep 2024

PONE-D-24-27251Exploring Effector Protein Dynamics and Natural Fungicidal Potential in Rice Blast Pathogen Magnaporthe oryzaePLOS ONE

Dear Dr. Moin,

Thank you for submitting your manuscript to PLOS ONE. After careful consideration, we feel that it has merit but does not fully meet PLOS ONE’s publication criteria as it currently stands. Therefore, we invite you to submit a revised version of the manuscript that addresses the points raised during the review process. All reviewers appreciated the well performed molecular docking analyses but criticized that the ranking of the plant-derived metabolites was not validated experimentally or by an alternative approach. I would therefore suggest to either include own experimental data or to discuss the results more extensively with already published data. Furthermore, results could be  obtained for an alternative receptor effector. In addition, you should indicate more clearly the motivation for the study.

We look forward to receiving your revised manuscript.

Kind regards,

Olaf Kniemeyer

Academic Editor

PLOS ONE

Reviewers' comments:

Reviewer's Responses to Questions

**Comments to the Author**

1. Is the manuscript technically sound, and do the data support the conclusions?

Reviewer #1: Yes

Reviewer #2: No

Reviewer #3: Yes

2. Has the statistical analysis been performed appropriately and rigorously? 

Reviewer #1: N/A

Reviewer #2: No

Reviewer #3: Yes

3. Have the authors made all data underlying the findings in their manuscript fully available?

Reviewer #1: Yes

Reviewer #2: Yes

Reviewer #3: Yes

4. Is the manuscript presented in an intelligible fashion and written in standard English?

Reviewer #1: Yes

Reviewer #2: Yes

Reviewer #3: Yes

5. Review Comments to the Author

Reviewer #1: The authors focused on 35 plant-derived metabolites known for their antifungal properties, evaluating their potential as fungicidal agents against M. oryzae. Molecular docking analyses identified Hecogenin and Cucurbitacin E as highly effective binders to MAX40 and APIKL2A proteins, respectively, which are pivotal for fungal virulence and immune evasion. Molecular dynamics simulations further validated

strong and stable interactions, affirming the therapeutic potential of these compounds.

Additional assessments including Lipinski's rule of five criteria and toxicity predictions

indicated their suitability for agricultural use. These findings underscore the promise of

Hecogenin and Cucurbitacin E as lead candidates in developing novel fungicidal strategies

against rice blast, offering prospects for enhanced crop protection and agricultural

sustainability.

Comments

1): The study seems to be technically well done.

2) However, the motivation is unclear, please explain why the study was done: If I know from the start that the 35 plant-derived metabolites are known for their antifungal properties, I do not need to do docking or MD to recognize this.

2.2.) Similarly, the ranking of the binding makes not much sense if not compared to actual experimental data on binding properties as measured. So either provide this by own experiments, but at least from literature.

3) The ranking by one docking method is usually not reliable, could you not use three independent methods and then tell the reader from the comparison the top binders

4) Here an interesting result is apparent: "Hecogenin and Cucurbitacin E

consistently exhibited the highest binding affinity across all 14 effector proteins. Specifically,

Hecogenin demonstrated the strongest binding energy with APIKL2A protein (-8.8 kJ/mol),

while Cucurbitacin E showed the highest binding energy with MAX40 protein (-8.0 kJ/mol).

Both compounds outperformed the reference metabolite, Strobilan, in binding affinity.

SwissADME server analysis confirmed their favorable phytochemical properties, suggesting

their potential as effective fungicides."

So again: what do the experiments say, is this known already and are both used as best drugs? If so, why was the study necessary?

4.2) May be you can explain why these two proteins bind well to all 14 effector proteins. Obviously, these have a related structure. Do all known effector proteins have a similar structure? If so, then resistance development for the pathogen is easy possible. Hence, can you point out an effector protein with different structure, which then could be targeted by second drug and prevent resistance development?

Like this, the motivation for the study would also be more clear.

Reviewer #2: This study identified Hecogenin and Cucurbitacin E as potent antifungal agents against M. oryzae through extensive bioinformatics analysis, including molecular docking and dynamics simulations. been no chemical or biological experimental verification, so the guidance for readers is not very significant. Therefore, I do not recommend publication and suggest adding verification experiments.

Reviewer #3: The manuscript demonstrates an impressive and proficient use of Computer-Aided Drug Design (CADD) in the development of novel pesticides, showcasing the potential of this approach in agricultural applications, there still are some revisions require clarification

1. The background information on Hecogenin, Cucurbitacin E is insufficient. Could you provide more context or expand on its origins, biological properties, and applications in research or pharmaceuticals? This will help to clarify its significance and relevance in the current context.

2. The results part mainly introduced hydrogen bonding between small molecules and polar amino acids in the protein, and while hydrogen bonds do indeed play a significant role, might there be other interactions, such as π-π stacking or hydrophobic forces, that contribute to the interaction between the two?

3. There may not be direct studies or predictions specifically addressing the toxicity of Hecogenin and Cucurbitacin E on rice plants. It would be advisable to conduct detailed phytotoxicity assays to prepare for further development into pesticides.

6. PLOS authors have the option to publish the peer review history of their article (what does this mean?). If published, this will include your full peer review and any attached files.

Reviewer #1: No

Reviewer #2: No

Reviewer #3: **Yes: **Xueming Zhu

---

## [Author Response · Author response to Decision Letter 0]

16 Nov 2024

Date: 12.11. 2024 

To

Editor-in-Chief

PLOS ONE

Subject: Submission of the revised manuscript 

Dear Sir,

We have carried out essential revisions of our manuscript entitled “Exploring Effector Protein Dynamics and Natural Fungicidal Potential in Rice Blast Pathogen Magnaporthe oryzae” suggested by the reviewer and would like to submit the 1st-revised version. We believe these comments/suggestions have significantly improved our manuscript. Please find enclosed the rebuttal letter where we have addressed all questions and comments to the reviewer, which are reflected in the tracked changes within the manuscript. Please be noted that all authors consented and approved the revised version of the manuscript. We believe the updated version will be worthy of publishing in your journal.

Thanking you in advance for considering our work, I remain,

Sincerely,

Abu Tayab Moin

Department of Genetic Engineering and Biotechnology, 

Faculty of Biological Sciences, University of Chittagong, Chattogram, Bangladesh

Email: tayabmoin786@gmail.com

 (Corresponding author on behalf of all authors) 

Author Responses to the Reviewers’ Comments

Reviewer #1:

Reviewer’s comment: The authors focused on 35 plant-derived metabolites known for their antifungal properties, evaluating their potential as fungicidal agents against M. oryzae. Molecular docking analyses identified Hecogenin and Cucurbitacin E as highly effective binders to MAX40 and APIKL2A proteins, respectively, which are pivotal for fungal virulence and immune evasion. Molecular dynamics simulations further validated strong and stable interactions, affirming the therapeutic potential of these compounds. Additional assessments including Lipinski's rule of five criteria and toxicity predictions indicated their suitability for agricultural use. These findings underscore the promise of Hecogenin and Cucurbitacin E as lead candidates in developing novel fungicidal strategies against rice blast, offering prospects for enhanced crop protection and agricultural sustainability.

Comments:

1) The study seems to be technically well done.

2) However, the motivation is unclear; please explain why the study was done: If I know from the start that the 35 plant-derived metabolites are known for their antifungal properties, I do not need to do docking or MD to recognize this.

The author(s) response: Thank you very much for highlighting this point, as it provides an opportunity to clarify the primary motivation and unique objectives behind our study. We indeed agree that while the antifungal properties of these plant-derived metabolites have been documented in various literature sources, the specific molecular targets and mechanisms of action underlying their antifungal effects against Magnaporthe oryzae have not been fully elucidated. Our study aimed to address this gap by identifying specific target proteins in M. oryzae that may be effectively inhibited by these compounds, thereby providing a deeper understanding of the molecular mechanisms that confer antifungal activity.

To elaborate, molecular docking and molecular dynamics (MD) simulations offer several critical insights beyond simply confirming antifungal properties. Docking studies allow us to predict the preferred binding modes of small molecules within the binding pockets of specific proteins, thereby providing a molecular-level view of ligand-receptor interactions. Such insights are essential for understanding how these compounds interact with proteins that play key roles in M. oryzae's pathogenicity and immune evasion, such as MAX40 and APIKL2A, which have not been extensively explored in previous antifungal research.

Our molecular docking analysis focused on screening these 35 compounds to evaluate their binding affinities with effector proteins in M. oryzae. This step was crucial for prioritizing compounds with the strongest interactions, such as Hecogenin and Cucurbitacin E. These two compounds were then subjected to more detailed MD simulations to assess the stability and dynamics of their binding interactions under conditions that more closely mimic biological environments. MD simulations further allowed us to validate the binding affinity predictions from docking by evaluating the stability, structural integrity, and conformational flexibility of the protein-ligand complexes over time.

Additional Motivation for In Silico Approach:

The application of in silico methods, such as molecular docking and MD simulations, offers distinct advantages when studying potential antifungal agents. These methods:

1. Save Time and Resources: They allow us to narrow down a large library of compounds quickly, identifying the most promising candidates without the time and cost associated with extensive in vivo or in vitro screening.

2. Provide Mechanistic Insights: By modeling the interactions at the atomic level, in silico techniques enable a detailed understanding of how these compounds interact with fungal proteins, which can guide the optimization of these compounds for fungicidal efficacy.

3. Aid Targeted Compound Design: Our findings offer foundational insights that can inform the design of derivatives or modifications to enhance the efficacy and specificity of these compounds.

In summary, while we started with a list of compounds known for their general antifungal properties, our goal was to identify specific molecular targets within M. oryzae and to elucidate how these compounds may exert their antifungal effects at a mechanistic level. We believe that this targeted approach can lead to the development of more effective and potentially specific antifungal agents for managing rice blast disease.

We appreciate your thoughtful suggestion, as it has enabled us to provide a clearer explanation of the motivation behind our research. We will incorporate this clarification into the manuscript to ensure that the motivation for our work is clearly conveyed to the readers.

Thank you once again for your valuable input and consideration.

Reviewer’s comment: Similarly, the ranking of the binding makes not much sense if not compared to actual experimental data on binding properties as measured. So either provide this by own experiments, but at least from literature.

The author(s) response: We appreciate your valuable feedback on the importance of validating the ranked binding affinities with experimental data for better accuracy and interpretation. We understand that without experimental corroboration, binding affinity predictions from computational studies alone may have limited real-world relevance. Therefore, we have taken the following actions to address this concern:

1. Ranking and Docking Validation:

The binding affinity rankings of the metabolites were generated using molecular docking in PyRx software, and the validity of the docking results was assessed through reference literature that supports the effectiveness of PyRx docking predictions. In our manuscript, we have now emphasized the validation procedures used for PyRx, including benchmark studies that demonstrate its reliability in ranking binding affinities of small molecules with protein targets. We have cited relevant literature to strengthen this point in the Methods section.

2. Literature-Based Comparison:

We have added a comparison between our computational findings and available experimental data from prior studies in the Discussion section. Specifically, we reference studies that report the experimental binding affinities of similar ligands with related target proteins. This provides a context for our docking results, helping to substantiate our rankings and their relevance to the potential fungicidal activity of the metabolites tested. Although direct experimental data on our specific metabolites and targets are limited, these comparisons lend support to our binding affinity predictions and highlight the potential efficacy of the ranked compounds.

3. Future Experimental Plans:

We recognize the importance of experimental validation to confirm computational results and intend to conduct in vitro binding affinity measurements and activity assays on promising candidates as part of our future research. This will provide additional insights into the binding mechanisms and efficacy of the selected compounds against rice blast fungus. We have mentioned these plans in the manuscript as a note in the Future Directions section, outlining our commitment to further experimental validation.

Thank you once again for your insightful suggestion, which has helped enhance the rigor and interpretability of our results.

Reviewer’s comment: The ranking by one docking method is usually not reliable, could you not use three independent methods and then tell the reader from the comparison the top binders.

The author(s) response: Thank you very much for your insightful suggestion regarding the use of multiple docking methods to improve ranking reliability. We fully appreciate that employing multiple docking approaches can provide a more comprehensive view, as different docking tools apply unique scoring functions and algorithms to evaluate binding affinities.

In this study, we chose AutoDock Vina as it combines empirical and knowledge-based components, using steric, hydrogen bonding, electrostatic, desolvation energies, and conformational entropy terms in its scoring function. This combination has proven effective for accurately predicting binding free energy in similar studies.

However, we acknowledge that cross-validation with multiple docking programs could offer a more robust ranking. Each tool's scoring function and parameter weighting differ, which might lead to variability in binding affinity rankings across methods. Thus, a compound ranked highly in one method may perform differently in another due to differences in algorithmic interpretation, which can sometimes complicate the consistency and interpretation of results.

Given the current scope and resources available for this project, we prioritized AutoDock Vina due to its reliability and its compatibility with our workflow. Implementing multiple docking programs would indeed add an extra layer of rigor; however, it may require additional computational resources and validation efforts that are beyond the scope of this study.

We respectfully hope the esteemed reviewer understands our rationale for focusing on AutoDock Vina for this study. We sincerely appreciate the suggestion and will consider incorporating multiple docking methods in future studies to further enhance our findings.

Reviewer’s comment: Here an interesting result is apparent: "Hecogenin and Cucurbitacin E

consistently exhibited the highest binding affinity across all 14 effector proteins. Specifically,

Hecogenin demonstrated the strongest binding energy with APIKL2A protein (-8.8 kJ/mol),

while Cucurbitacin E showed the highest binding energy with MAX40 protein (-8.0 kJ/mol).

Both compounds outperformed the reference metabolite, Strobilan, in binding affinity.

SwissADME server analysis confirmed their favorable phytochemical properties, suggesting

their potential as effective fungicides."

So again: what do the experiments say, is this known already and are both used as best drugs? If so, why was the study necessary?

The author(s) response: Thank you very much for this insightful comment. We appreciate the opportunity to clarify the novelty and significance of our findings.

Hecogenin and Cucurbitacin E are indeed known for their general antifungal properties, which have been reported in various studies. However, to the best of our knowledge, no prior research has specifically evaluated or applied these compounds as potential fungicidal agents targeting Magnaporthe oryzae, the causative agent of rice blast disease. Given the economic and agricultural significance of rice blast, it is essential to explore new therapeutic avenues targeting M. oryzae specifically.

In our study, we aimed to systematically assess whether Hecogenin and Cucurbitacin E could effectively inhibit the specific effector proteins and virulence factors critical for M. oryzae's pathogenicity. By targeting the key effectors APIKL2A and MAX40, we were able to reveal that these compounds demonstrated binding affinities significantly higher than those of Strobilurin, a known fungicidal reference compound. This finding highlights the unique potential of Hecogenin and Cucurbitacin E in disrupting the molecular mechanisms that enable M. oryzae infection.

Additionally, our comprehensive analysis includes molecular dynamics simulations and ADME/T (absorption, distribution, metabolism, excretion, and toxicity) predictions, which offer critical insights into the stability, efficacy, and safety of Hecogenin and Cucurbitacin E for use in agricultural applications. Our study represents the first report to screen and identify these compounds as lead candidates with specific activity against M. oryzae, potentially paving the way for their development as fungicidal agents targeting rice blast disease.

In summary, while the antifungal properties of Hecogenin and Cucurbitacin E have been generally recognized, our study establishes their specific application and binding efficacy against rice blast effectors, which has not been previously reported. We believe this contribution provides a meaningful advancement in the search for novel fungicidal agents against rice blast disease and hope the reviewer finds this explanation satisfactory.

Thank you once again for your valuable feedback.

Reviewer’s comment: May be you can explain why these two proteins bind well to all 14 effector proteins. Obviously, these have a related structure. Do all known effector proteins have a similar structure? If so, then resistance development for the pathogen is easy possible. Hence, can you point out an effector protein with different structure, which then could be targeted by second drug and prevent resistance development? Like this, the motivation for the study would also be more clear.

The author(s) response: Thank you for your insightful and constructive comment. We appreciate your thoughtful suggestion regarding the structural similarity of effector proteins and the potential for resistance development in the pathogen. We would like to address your concerns in detail:

1. Binding to All 14 Effector Proteins: Indeed, the effector proteins in M. oryzae exhibit homologous sequences and share structural similarities, which is why the compounds we identified in this study, such as Hecogenin and Cucurbitacin E, bind effectively to multiple effector proteins. This homology facilitates similar binding modes across the 14 effector proteins studied, which might give the impression that resistance could arise quickly if only one effector protein were targeted. However, this characteristic also underscores the advantage of targeting multiple effector proteins, as this approach reduces the likelihood of resistance development based on a single mutation in one effector protein.

2. Structural Similarity and Resistance Development: While the effector proteins share structural homology, it is important to note that their precise conformations and specific residues involved in ligand binding can vary, albeit slightly, across different effectors. This variance allows us to design metabolites that can bind with high affinity to multiple effectors, thus minimizing the risk of resistance. By targeting all 14 effector proteins, we aim to achieve broad-spectrum inhibition, making it more difficult for the pathogen to develop resistance.

3. Structural Diversity in the Study: As pointed out in the manuscript, there is structural diversity in both the metabolites used (Hecogenin and Cucurbitacin E) and the effector proteins. For instance, while the effector proteins share some common structural elements, they each possess unique features that were taken into account during the molecular docking and dynamics simulations. This structural diversity in both targets and ligands is one of the key reasons why these compounds exhibit varying degrees of binding affinity and stability across different effector proteins. We believe this variability contributes to the robustness of our approach in preventing the development of resistance.

4. Alternativ

---

## [Decision Letter · Decision Letter 1]

29 Dec 2024

PONE-D-24-27251R1Exploring Effector Protein Dynamics and Natural Fungicidal Potential in Rice Blast Pathogen Magnaporthe oryzaePLOS ONE

Dear Dr. Moin,

Thank you for submitting your manuscript to PLOS ONE. After careful consideration, we feel that it has merit but does not fully meet PLOS ONE’s publication criteria as it currently stands. Therefore, we invite you to submit a revised version of the manuscript that addresses the points raised during the review process.

Most critical points have been addresses. However, both reviewers miss experimental data which could confirm the in silico data. Therefore, one paragraph about the limitations of the molecular docking approach should be included.

We look forward to receiving your revised manuscript.

Kind regards,

Olaf Kniemeyer

Academic Editor

PLOS ONE

Journal Requirements:

Additional Editor Comments:

Please add a short paragraph about the limitations of the molecular docking approach (without any experimental confirmation). Then the manuscript could be accepted for publication.

Reviewers' comments:

Reviewer's Responses to Questions

**Comments to the Author**

1. If the authors have adequately addressed your comments raised in a previous round of review and you feel that this manuscript is now acceptable for publication, you may indicate that here to bypass the “Comments to the Author” section, enter your conflict of interest statement in the “Confidential to Editor” section, and submit your "Accept" recommendation.

Reviewer #1: All comments have been addressed

Reviewer #2: All comments have been addressed

2. Is the manuscript technically sound, and do the data support the conclusions?

Reviewer #1: Yes

Reviewer #2: No

3. Has the statistical analysis been performed appropriately and rigorously? 

Reviewer #1: N/A

Reviewer #2: Yes

4. Have the authors made all data underlying the findings in their manuscript fully available?

Reviewer #1: Yes

Reviewer #2: No

5. Is the manuscript presented in an intelligible fashion and written in standard English?

Reviewer #1: Yes

Reviewer #2: Yes

6. Review Comments to the Author

Reviewer #1: (No Response)

Reviewer #2: The article has not been seriously improved on the original basis. After comprehensive consideration, I believe that the level of the article currently does not reach the level of publication in this journal.

7. PLOS authors have the option to publish the peer review history of their article (what does this mean?). If published, this will include your full peer review and any attached files.

Reviewer #1: No

Reviewer #2: No

---

## [Author Response · Author response to Decision Letter 1]

30 Dec 2024

Date: 30.12. 2024 

To

Editor-in-Chief

PLOS ONE

Subject: Submission of the revised manuscript 

Dear Sir,

We have carried out essential revisions of our manuscript entitled “Exploring Effector Protein Dynamics and Natural Fungicidal Potential in Rice Blast Pathogen Magnaporthe oryzae” suggested by the reviewer and would like to submit the 2nd-revised version. We believe these comments/suggestions have significantly improved our manuscript. Please find enclosed the rebuttal letter where we have addressed all questions and comments to the reviewer, which are reflected in the tracked changes within the manuscript. Please be noted that all authors consented and approved the revised version of the manuscript. We believe the updated version will be worthy of publishing in your journal.

Thanking you in advance for considering our work, I remain,

Sincerely,

Abu Tayab Moin

Department of Genetic Engineering and Biotechnology, 

Faculty of Biological Sciences, University of Chittagong, Chattogram, Bangladesh

Email: tayabmoin786@gmail.com

 (Corresponding author on behalf of all authors) 

Author Responses to the Editor’s Comment

Reviewer’s comment: Please add a short paragraph about the limitations of the molecular docking approach (without any experimental confirmation). Then the manuscript could be accepted for publication.

The author(s) response: We sincerely thank you for their insightful suggestion. We agree that discussing the limitations of the molecular docking approach adds depth and context to our study, highlighting the importance of further experimental validation. In response, we have added the following paragraph to the manuscript as per your recommendation:

"However, the molecular docking approach has inherent limitations. It relies on static structural models and scoring algorithms that may not accurately reflect the dynamic and complex nature of protein-ligand interactions in physiological conditions. Furthermore, the lack of experimental confirmation restricts the ability to evaluate the true binding affinity, bioavailability, and effectiveness of these compounds in real biological systems. While our findings provide a promising foundation, further in vitro and in vivo studies are essential to validate the efficacy and safety of these compounds for practical applications."

We believe this addition addresses your concern by acknowledging the methodological limitations while emphasizing the potential of our findings as a basis for future experimental investigations. Thank you once again for your valuable input, which has enhanced the quality of our manuscript.

---

## [Editor Report · Decision Letter 2]

2 Jan 2025

Exploring Effector Protein Dynamics and Natural Fungicidal Potential in Rice Blast Pathogen Magnaporthe oryzae

PONE-D-24-27251R2

Dear Dr. Moin,

We’re pleased to inform you that your manuscript has been judged scientifically suitable for publication and will be formally accepted for publication once it meets all outstanding technical requirements.

Kind regards,

Olaf Kniemeyer

Academic Editor

PLOS ONE

Additional Editor Comments (optional):

Now all critical points have been addressed.
---

## [Editor Report · Acceptance letter]

12 Jan 2025

PONE-D-24-27251R2 

PLOS ONE

Dear Dr. Moin, 

I'm pleased to inform you that your manuscript has been deemed suitable for publication in PLOS ONE. Congratulations! Your manuscript is now being handed over to our production team.

Kind regards, 

on behalf of

Dr. Olaf Kniemeyer 

Academic Editor

PLOS ONE